# Hippocampus supports multi-task reinforcement learning under partial observability

Dabal Pedamonti[1,2,6], Samia Mohinta [1,3,6], Martin V. Dimitrov[1], Hugo Malagon-Vina [4], Stephane Ciocchi [4,5] & Rui Ponte Costa [1,2,5] ✉

Mastering navigation in environments with limited visibility is crucial for survival. Although the hippocampus has been associated with goal-oriented navigation, its role in real-world behaviour remains unclear. To investigate this, we combined deep reinforcement learning (RL) modelling with behavioural and neural data analysis. First, we trained RL agents in partially observable environments using egocentric and allocentric tasks. We show that agents equipped with recurrent hippocampal circuitry, but not purely feedforward networks, learned the tasks in line with animal behaviour. Next, we used dimensionality reduction of the agents' internal representations to extract components reflecting reward, strategy, and temporal representations, which we validated experimentally against hippocampal recordings from rats. Moreover, hippocampal RL agents predicted state-specific trajectories, mirroring empirical findings. In contrast, agents trained in fully observable environments failed to capture experimental observations. Finally, we show that hippocampal-like RL agents demonstrated improved generalisation across novel task conditions. In summary, our findings suggest an important role of hippocampal networks in facilitating reinforcement learning in naturalistic environments.

As we navigate new environments, we must learn to integrate incomplete sensory information towards desired goals. How biological neural networks perform this feat is not fully understood.

The hippocampus is classically associated with the building of a cognitive map of the environment and the storage of episodic memories[1–3]. However, growing evidence suggests that the hippocampus also supports goal-driven behaviour[4–8]. For example,[4] showed that the hippocampus is indeed involved in planning routes towards desired goals. Moreover, their work suggests that hippocampal sequence events, known as "replay", serve as a mechanism for goal-directed navigation, facilitating memory-based trajectory planning and guiding subsequent navigational behaviour. Other studies have shown that the hippocampus, and the hippocampal recurrence, such as that provided by CA3 and CA1-EC loops, is involved in integrating information over time as required for navigational tasks where sensory cues are no longer present[9–14]. Given that in most naturalistic conditions animals do not have continuous access to the full environment, we postulate that the hippocampus may have evolved to support goal-driven navigation in environments in which sensory information is not always present.

The hippocampus has been traditionally conceptualised using Hopfield neural networks, known for their capacity for autoassociative

[1]Computational Neuroscience Unit, Intelligent Systems Labs, Faculty of Engineering, University of Bristol, Bristol, UK. [2]Centre for Neural Circuits and Behaviour, Department of Physiology, Anatomy and Genetics, University of Oxford, Oxford, UK. [3]Department of Physiology, Development and Neuroscience, University of Cambridge, Cambridge, UK. [4]Center for Brain Research, Division of Cognitive Neurobiology, Medical University of Vienna, Vienna, Austria. [5]Laboratory of Systems Neuroscience, Department of Physiology, University of Bern, Bern, Switzerland. [6]These authors contributed equally: Dabal Pedamonti, Samia Mohinta. ✉e-mail: rui.costa@dpag.ox.ac.uk

memory storage[15,16]. More recent studies have demonstrated that recurrent neural network models of the hippocampus, trained to predict next state or position, exhibit specific cell types tuned to spatial information[17–21], including the commonly observed place cells and grid cells[22,23]. Another line of research has shown that some model-based approaches, such as Successor Representation, can explain anticipatory features that have been observed experimentally[24,25]. A related approach combined model-free with model-based networks of the hippocampal-striatal system to reproduce a range of behavioural findings from spatial and non-spatial decision-making tasks[26].

A complementary viewpoint has explored the involvement of the hippocampus in reward-centric navigational tasks[6,7,27–30]. For instance,[27,29] utilised an actor-critic network coupled with goal-independent representations to model water-maze tasks[31], on the other hand, proposed a model that integrates unsupervised learning for spatial modelling with reward-based learning for goal-oriented behaviour. Nevertheless, such models did not explicitly differentiate the functional impact of hippocampal recurrent networks. Consequently, the contribution of the classical hippocampal structure for navigation of goal-driven environments under partially observable conditions, the underlying neural dynamics, and its implications for animal behaviour remain unclear.

Here, we posit that the hippocampal circuitry, particularly hippocampal recurrence, such as observed in CA3 recurrence, is adept at navigating environments under realistic conditions, particularly those characterised by limited visibility and cue uncertainty. To test this hypothesis, we combine behavioural and neural data analysis together with deep reinforcement learning (RL) modelling of goal-driven tasks. Both animals and agents were trained to perform ego-allocentric strategies on a T-maze. Our models consist of a network trained end-to-end to perform goal-driven tasks using deep reinforcement learning. By contrasting experimental observations with the model we show that hippocampal networks trained in partial, but not fully observable environments, provide a good match of neuronal and behavioural observations. Using task-relevant dimensionality reduction, we show that hippocampal neurons encode decision, strategy, and temporal population activity, which are better captured by a model incorporating recurrence. Moreover, our modelling shows that recurrence also captures key behavioural features commonly observed in animals and humans, and that it generalises to different task conditions. This is in contrast with non-recurrent models, which failed to capture experimental observations. In addition, our work shows that agents trained in fully observable environments also do not capture experimental observations, thus suggesting the need to reevaluate previous experimental findings that may have implicitly assumed full observability.

## Results

We were inspired by a standard behavioural setup in which animals are trained on plus-mazes[5]. Next, we highlight the key elements of this experimental setup, which we model. First, rats were trained in a multi-task setting in which they had to perform a goal-driven navigational task while following two strategies, egocentric and allocentric (Fig. 1a). In the egocentric (self-centred) rule, the reward was always positioned

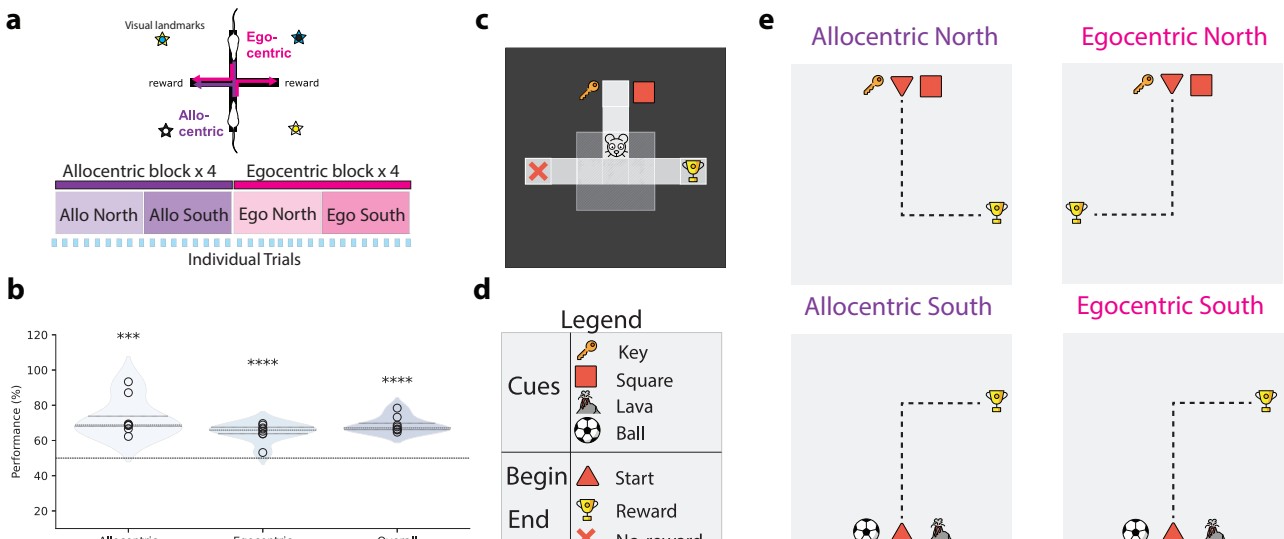

**Fig. 1 | Ego-allocentric task setup in animals and reinforcement learning agents. a** Top, experimental setup in which rats were placed in a plus-shaped maze, which was effectively transformed into a T-maze on each trial by blocking the opposite arm[5]. The task consists of reaching the reward at the end of one of two arms following either egocentric (pink) or allocentric (purple) rules. Bottom: both animal and artificial agents were trained by interleaving blocks of allocentric and egocentric (see main text). **b** Animal performance on allocentric and egocentric tasks following the setup shown in (**a**) ($n =8$)[5,57]. There are no statistically significant differences in performance between allocentric and egocentric perspectives. Horizontal dashed line represents the chance level. Data distribution is represented as a violin plot ($p_{allo} = 5.55 \times 10^{-4}$, $p_{ego} = 8.31 \times 10^{-5}$ and $p_{overall} = 8.02 \times 10^{-6}$). Each distribution is compared to the change level (50%). ***: $p < 0.001$, ****: $p < 0.0001$ (one-sample, two-sided t-test against chance level). **c** The experimental setup in (**a**) was simulated using a grid world environment (see Supplementary Movies). This setup was then used to train reinforcement learning agents in ego- and allocentric tasks. Environment observability was modelled by defining a visible range around the animal (light grey box), which is limited to the current cell alone when the agent

enters the terminal arms. Specifically, the agent can only see the squares directly in front of it and to its sides, but not behind it. A total of four cues (cf. (**d**)) are placed in the environment, two for the north starting state and two for the south starting state. The trophy and red cross represent rewarding and non-rewarding terminal states, respectively (not made visible to the agent). Schematics showing the cues used (**d**) and the four possible allocentric and egocentric rules (**e**). The dotted line represents the ideal path towards the reward, starting north/south. Cues are presented near the starting positions where key/square refer to the north starting point and lava/ball to the south one. In the allocentric task, the reward is always on the same side regardless of the starting position. In the egocentric task, the reward is always on the right side of the starting position. Source data are provided as a Source Data file. **a** was adapted from S Ciocchi, J Passecker, H Malagon-Vina, N Mikus, T Klausberger. Selective information routing by ventral hippocampal CA1 projection neurons. Science 348 (2015); reprinted with permission from AAAS. Icons used in panels **c**, **d**, and **e** are released by OpenMOJI under a Creative Commons Attribution ShareAlike license 4.0.

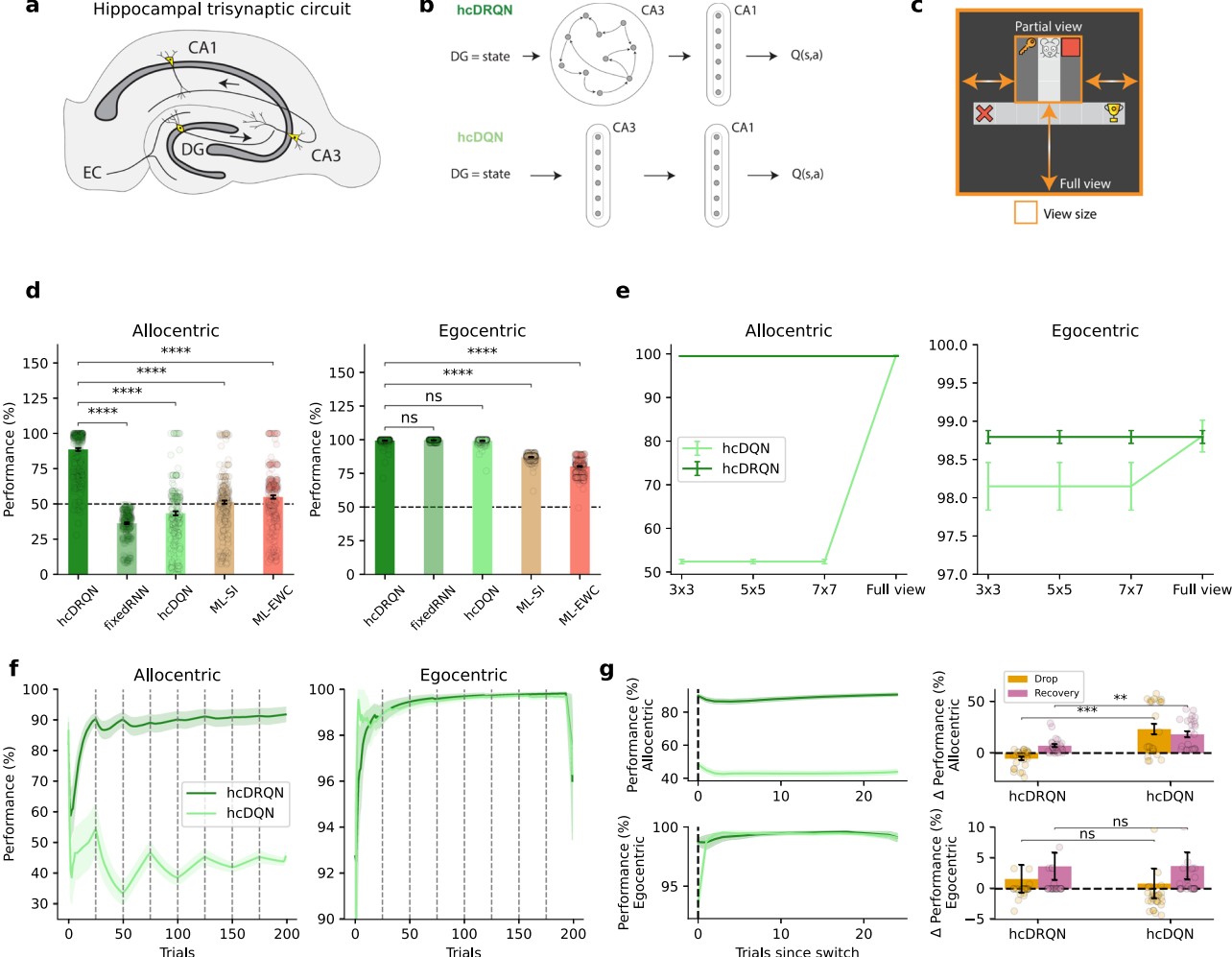

**Fig. 2 | Reinforcement learning agents with recurrence jointly learn ego and allocentric tasks. a** Classical hippocampal trisynaptic circuitry: entorhinal cortex (EC), dentate gyrus (DG), and hippocampus CA3 and CA1 layers. **b** Schematics of reinforcement learning (RL) agents with hippocampal-like architecture modelled as deep-Q-networks (DQN) used to learn the goal-driven tasks described in Fig. 1. In our models, the DG receives a simplified (partially observable) map of the environment, which is processed by the CA3-CA1 pathway and then CA1 projects to the reward system to compute a Q-value of state-action pairs, Q(s,a). We consider two main models: (i) with recurrence (hcDRQN, top) or (ii) with CA3 as a feedforward network (hcDQN, bottom). Both models consist of two hidden layers (CA3 and CA1) and an output action readout layer. **c** Minigrid environment showing 3 × 3 and full view size (orange outline). **d** Performance of all models for allocentric (left), egocentric (middle) tasks. For comparison with modern machine learning solutions to multi-task learning we also consider two popular algorithms: elastic weight consolidation (ML-EWC) and synaptic intelligence (ML-SI). Data are presented as mean values ± SEM ($n = 240$). $p\_allo$ (bottom to top) = $6.844e^{-167}$, $2.304e^{-92}$, $2.273e^{-83}$, $1.293e^{-69}$. $p\_ego$ (bottom to top) = $8.004e^{-02}$, $3.616e^{-01}$, $1.940e^{-224}$, $4.395e^{-183}$. **e** Task performance of RL agents as environment observability is progressively incremented. Both models achieve the same performance under full observability, whereas only the hcDRQN agent can learn tasks under non-full observability. Data are presented as mean values ± SEM over 5 different initial conditions. **f**, Learning curves for both hcDQN and hcDRQN, showing that the former fails to learn allocentric tasks. Vertical dashed lines represent switching points. Data are presented as mean values ± SEM over 5 different initial conditions. **g** Task performance aligned to task point of the task switch (left). Performance drop and recovery after task switching for both hcDRQN and hcDQN (right). Data are presented as mean values ± SEM ($n = 25$). $p_{allo,drop} = 1.18 \times 10^{-5}$, $p_{allo,rec} = 2.71 \times 10^{-3}$, $p_{ego,drop} = 0.431$, $p_{ego,rec} = 0.903$. **: $p < 0.01$, ***: $p < 0.001$, ****: $p < 0.0001$, ns indicates no significant (independent two-sample, two-sided t-tests across models). Icons used in panels c are released by OpenMOJI under a Creative Commons Attribution ShareAlike license 4.0. Source data are provided as a Source Data file.

in the same location with respect to the animal, i.e., regardless of the north or south starting location. For the allocentric (world-centred) rule, the reward is at the same location irrespective of the animal's starting position, and the animal needs to turn left or right depending on whether they start from a north or south position. In a given block, rodents have to infer the task (ego versus allocentric) through trial-and-error. Second, rats were placed on a standard plus-maze defined by a 5cm wall with cues in the surrounding walls. In such setups – and in most navigational task settings – rats are unlikely to have access to all information necessary for decision-making, implying a degree of partial observability[32]. Furthermore, we analysed the speed and head direction of rats as they navigated the maze. This analysis suggested

that rats did not have constant access to the full state of the environment (Fig. S1). Note that the maze setup was effectively transformed into a T-maze on each trial by blocking the opposite starting arm (Fig. 1a). Finally, animals were trained in a block-wise fashion with interleaved blocks containing allocentric and egocentric tasks (Fig. 1a). Each block contains sub-blocks corresponding to different starting locations (i.e., north versus south). Despite the relatively complex nature of these tasks with multiple rules, rats can learn all tasks (Fig. 1b).

Building on this task setup, we aimed to study the underlying architectural principles that enable such goal-driven navigation. To this end, we contrast animal behaviour and (hippocampal) neural data

with artificial reinforcement learning (RL) agents equipped with a hippocampal-like architecture. To mimic the experimental setup described above, we simulated a 2D minigrid discrete environment[33], which consists of a starting state and two terminal states: rewarded and non-rewarded (Fig. 1c). To capture both north and south starting states, we use different sensory cues (Fig. 1d). Closely following the experimental setup described above, we train the model in a multi-task (allocentric north/south and egocentric north/south) using the same block-wise training setup, as illustrated in Fig. 1e. As in the experimental setting, RL agents need to figure out the task context through trial-and-error, but this computation in the model is simplified by assuming separate ego and allocentric output action networks (see Methods). We provide a more detailed comparison between our modelling and the experimental setup in Table S1. Motivated by the fact that animals cannot continuously access all sensory information available in their environment (e.g., due to poor visual acuity and head direction), we test different degrees of observability of the environment.

## Agents with recurrence learn ego-allocentric goal-driven tasks

Our hippocampal RL models are based on standard deep RL models, specifically, Deep Q-networks (DQN)[34] (see Methods). Our models feature a three-layer hippocampal-like structure: the input layer emulates entorhinal input to the Dentate Gyrus (DG), the first layer represents CA3 and the second CA1. In addition, we also consider an output layer that encodes action-state values, denoted as $Q(a, s)$ (Fig. 2a, b). Environmental cues are provided around the start location of the maze and agents can take one of three actions (move forward, rotate left or rotate right). The entorhinal cortex (EC) is known to supply the hippocampus with a spatial map of the environment[23], which we approximate by providing our model with a 2D top-down spatial map obtained from a minigrid environment (Fig. 1c). The output layer, which encodes state-action Q-values, provides an abstraction of hippocampal-to-striatal networks[35]. To prevent forgetting of a previously learned task, we use two separate output Q-heads in all models. Consequently, we switch between the allocentric or egocentric head depending on the current task being performed by the agent. However, other strategies are possible, we also demonstrate that the same tasks can be learned using a contextual task-specific signal (task ID; Fig. S2), as is commonly done in computational neuroscience[36].

Motivated by the existence of recurrent connectivity in CA3[37] and in line with previous work in which brain areas are modelled as gated recurrent neural networks[38,39] we model CA3 using a Gated Recurrent Unit (GRU) network[40,41], which we denote as hippocampal deep recurrent Q-Network (hcDRQN). In addition, we contrast this network with three other networks: a purely feedforward hippocampal Deep Q-Network (hcDQN) and two hcDQNs augmented with artificial continual learning algorithms. We considered two modern machine learning algorithms as a strong control for what is achievable with feedforward networks in our multi-task setup. In particular, we included two of the most popular methods: Elastic Weight Consolidation (ML-EWC;[42]) and Synaptic Intelligence (ML-SI;[43]). Unlike standard DQN implementations, our setting did not require experience replay[34]. We therefore simplified our model by omitting this component. While experience replay is often interpreted in terms of hippocampal-cortical systems consolidation, there is also evidence for internal hippocampal replay[44,45]. Replay mechanisms are likely essential for stable reinforcement learning in more complex tasks than the ones that we consider[34].

First, we compare the hippocampal deep RL models (Fig. 2b) with animals by contrasting their task performance in a partially observable setting (Fig. 1b). We trained the models using the grid environment described above and following a similar training procedure (trial-by-trial) used to train animals with blocks of allocentric trials alternated with blocks of egocentric trials (Fig. 1a, bottom). Within each block we

alternate the two starting (north/south) positions. Our results show that the network with recurrence, hcDRQN, can successfully learn multiple tasks (Fig. 2d). Moreover, hcDRQN is the only model that can learn allocentric tasks while other models perform around chance level. These results show that hcDRQN learns sufficiently generalisable representations across all tasks considered here. This is due to the RNN's ability to integrate information over time, thus being able to link sensory cues with future outcomes. The action-value outputs must then decode this mixed information from the hippocampal network to attribute value to the appropriate actions. In contrast, hcDQN only learns a subset of the tasks. This is because models that fail to truly learn the tasks will default to memorising to always turn right at the decision point as this behaviour will work on 3 of the 4 sub-tasks (allo-south, ego-north and ego-south). This is in line with the performance observed experimentally, showing that animals can learn both strategies (Fig. 1b). In addition, our results show that a model without plasticity in the recurrent CA3 layer is not sufficient to learn all tasks (fixed hcDRQN model), which further supports the importance of the RNN for learning these tasks. Studying how the performance evolves over trials within each allocentric and egocentric block, shows that allocentric performance drops for hcDQN after each switch between north vs. south scenarios (Fig. 2f). Performance aligned to allocentric goal switches shows that hcDRQN has small delta values for the drops and recovery, reflecting lower errors to changes in goal location (Fig. 2g, top row). hcDQN, by contrast, shows larger delta values, indicating a failure to form or update allocentric representations. Comparing hcDRQN vs hcDQN, both the drop and recovery effects remained significant ($p\_drop$ = 1.18e-5, $p\_rec$ = 2.71e-3) in the allocentric case, confirming that hcDRQN learns and adapts better to the allocentric subtask. Drop is defined as the difference between the mean performance of the first block and the minimum of the remaining blocks, and recovery as the difference between the mean of the last two blocks (plateau values) and the minimum calculated in the previous drop. No such effects are observed in the egocentric task (Fig. 2g, bottom row). Finally, machine continual learning models perform similarly to hcDQN, thus not being able to learn all tasks (Fig. S3c).

## Recurrence is needed in partially observable environments

Hippocampal recurrence, such as provided by CA3 in our hcDRQN agent enables information to be integrated over time, allowing it to retain initial cues for accurate action selection at decision points. Therefore, the hcDQN model should match hcDRQN performance in fully observable environments. To demonstrate this, we tested hcDRQN and hcDQN in environments with different degrees of visibility ($3 \times 3$, $5 \times 5$, $7 \times 7$, and full view; Fig. 2c). We expected a model without recurrence (i.e., DQN) to be able to solve all tasks in environments with full observability (i.e., all information continuously available). Our results show that the non-recurrent model, hcDQN, only succeeds to learn both allo and egocentric tasks when the full view is provided (Fig. 2e). We expected that models learn to solve the task in these conditions by continuously relying on having access to the task-specific cues. To test this continuous reliance on sensory cues we removed the cues after the decision point (Fig. S4). Our results show that both models completely fail to complete the tasks.

Given that animals are unlikely to have continuous access to the full environment in most realistic settings, our results suggest that hippocampal recurrence—such as that provided by CA3, though not exclusively—plays an important role in supporting goal-directed behaviour under naturalistic conditions.

## Agent's task-relevant neuronal dynamics are in alignment with experimental recordings

Next, we tested whether RL agents trained in partially observable environments best capture experimental task-specific neural dynamics. To this end, we contrasted the neural dynamics predicted by the

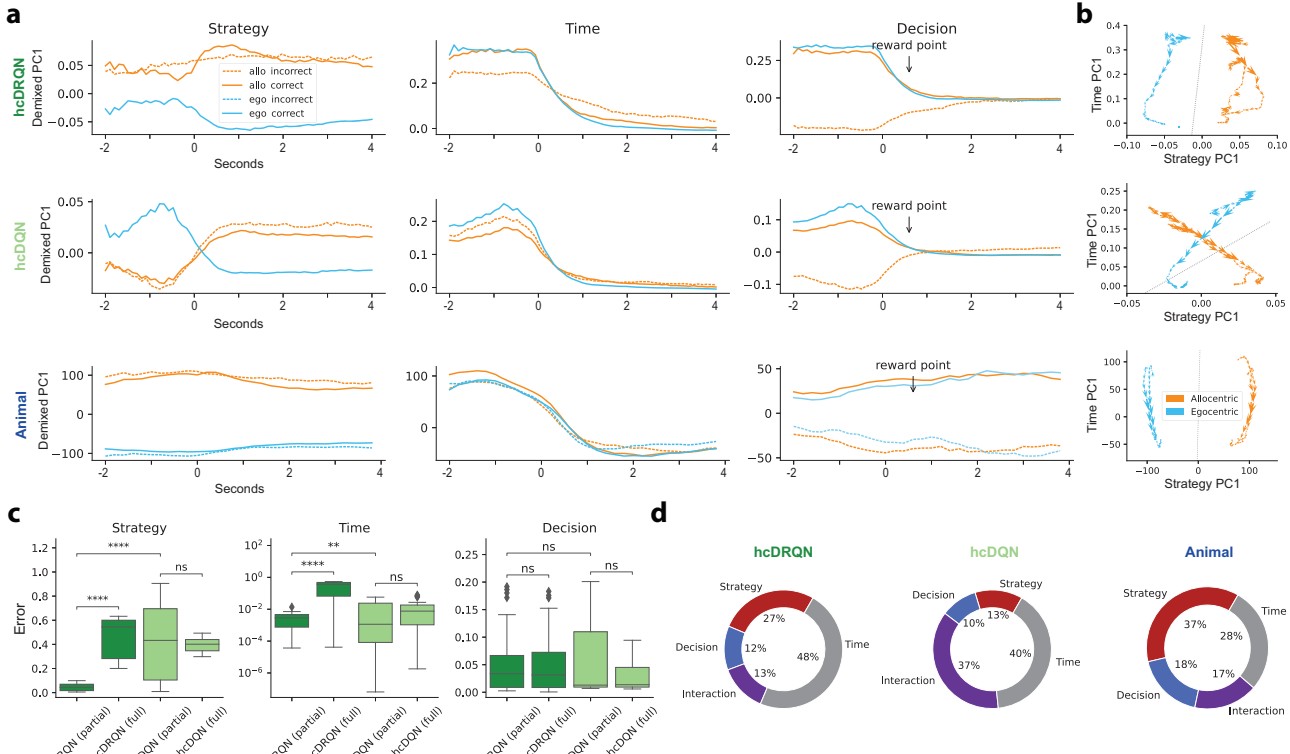

**Fig. 3 | Strategy, temporal, and outcome neural dynamics in RL agents and animals. a** Demixed principal components corresponding to task strategy (allocentric and egocentric for both north and south start locations), time, and decision (correct and incorrect). hcDRQN components show separate task strategies, whereas hcDQN mixes task strategies. Note that RL agents do not show any egocentric incorrect dynamics because they learn these tasks perfectly (i.e., without incorrect trials). All components are provided in Figs. S6, S7 and S8. **b** Strategy and time demixed components colour-coded by strategy (allocentric in orange and egocentric in blue). Dashed lines indicate the decision boundary obtained by a linear classifier. Arrows denote the direction of progression over time. **c** Mean squared error between normalised model and the animal data demixed components. We also contrast agents trained with full and partial observability (cf. Fig. S9).

Box plots show median (center line), interquartile range (box = Q1-Q3), and whiskers extending to the most extreme data points within 1.5 × IQR from Q1/Q3; outliers beyond this range are not displayed. Independent two-sample, two-sided t-tests across models ($n = 30$). **: $p < 0.01$, ***: $p < 0.001$, ****: $p < 0.0001$, ns indicates no significant. $p$-values strategy: $p_{hcDRQN(partial) vs full} = 4.04 \times 10^{-20}$, $p_{hcDQN(partial) vs full} = 0.763$, $p_{hcDRQN vs hcDQN (partial)} = 8.58 \times 10^{-9}$; time: $p_{hcDRQN(partial) vs full} = 1.62 \times 10^{-9}$, $p_{hcDQN vs hcDQN (full)} = 0.448$, $p_{hcDRQN(partial) vs hcDQN (partial)} = 8.05 \times 10^{-3}$; decision: $p_{hcDRQN(partial) vs full} = 0.887$, $p_{hcDQN(partial) vs full} = 0.0616$, $p_{hcDRQN vs hcDQN (partial)} = 0.834$. **d** Percentage of explained variance for all combined Decision, Strategy, Interaction and Time components (cf. with individual components in Fig. S6). Source data are provided as a Source Data file.

model with those recorded from awake performing animals by using dimensionality reduction techniques. In particular, we used demixed PCA (dPCA), which enabled us to extract behaviourally-relevant dimensions.

We extracted task-specific neural activity from agents in an intermediate stage of learning as to have a more comparable performance to that observed in animals (cf. Fig. 1b and Fig. S5a). The simulated neural activity of the modelled CA1 layer was then used to perform dPCA (see Methods for details). This analysis focused on three task-encoding components of interest in the hcDRQN agent (Fig. 3a): strategy, time, and decision. First, we find that the population dynamics for allo and egocentric tasks can be linearly separable when using the hcDRQN strategy demixed PCs, but not for hcDQN (Fig. 3a left, Fig. 3b). This reversal between tasks by hcDQN is likely due to its inability to learn both allocentric and egocentric tasks (Fig. 1). The fact that hcDRQN, but not hcDQN, yields a clear separation of strategies is consistent with our results above showing that it can learn both allo and egocentric strategies (Fig. 2d). Next, we find that both hcDRQN and hcDQN exhibit a temporally decaying population dynamics (Fig. 3a middle). However, whereas the hcDRQN starts flat and then decays, the hcDQN increases before it decays. These hcDRQN neural dynamics are more in line with the animal neural dynamics (cf. Fig. 3c). Finally, we observe decision- or outcome-specific components (correct versus incorrect), which, as expected, collapse once the reward

terminal point is reached. This predicts that the hippocampus should encode reward/no-reward information well before the terminal states. When sensory cues are removed, these task-specific features in the strategy and decision dPCA components are eliminated, aligning with the role of RNN sensory integration in achieving correct task outcomes (Fig. S10).

Next, to contrast the neural representations predicted by the RL agents, we used tetrode recordings obtained from 612 CA1 neurons, as described in ref. 5. To conduct this analysis, neural data were aligned to the moment when the reward zone was entered (see Methods). The results of the dPCA show that the data qualitatively validate the results predicted by the hcDRQN agent for the strategy-specific components, but not the hcDQN agent (Fig. 3a bottom). We also observe stronger neuronal activations in the allocentric tasks compared to the egocentric tasks, in line with experimental observations (Fig. S11). To better quantify model-data match we used a normalised mean-squared error metric (see Methods). This metric shows that, indeed, hcDRQN better captures experimentally observed strategy neural dynamics (Fig. 3c left). For the time-specific components, we observe a decaying component as predicted by the RL agents. Although both the hcDRQN and hcDQN look qualitatively very similar, the error metric shows that hcDRQN provides a better match with the data (Fig. 3c middle). Finally, the decision component also reveals a separation between correct and incorrect trials as predicted by the models, but in contrast to the

models, this separation remains after the reward point. This discrepancy between model and animal reward-relevant neural dynamics is likely rooted in the distinction between reinforcement learning agents, which receive instantaneous rewards, and animals, which require several seconds to consume rewards. However, we should point out that reward points in experimental data simply mean that a sensor close to the reward was triggered, thus there is likely some delay between triggering the sensor and actually perceiving the reward. As before, we used the model-data error metric on the decision components (up to the reward point) and found no differences between the hcDQN and hcDRQN (Fig. 3c right).

Models trained under full observability conditions do not appear to provide a good match with experimental observations. To further test this point, we compared model neural dynamics in agents trained with full continuous access to the environment (Fig. S9). Our error metric shows that agents trained with full observability provide a poor match to neural dynamics when compared to agents trained under partial observability (Fig. 3c). These results provide further support for hippocampal recurrence as being important to navigate environments under more naturalistic conditions, such as partial observability.

Finally, we contrast the degree of explained variance across models and data. hcDRQN captures explained variance across behavioural variables in a way that more closely matches awake tetrode recordings (Fig. 3d). Of particular interest is the fact that hcDQN relies more on mixed (or interaction) components (37%) compared to hcDRQN (13%) and animal (17%), which is in line with its inability to fully solve all the tasks. On the other hand, for both hcDRQN and animal dynamics, but not hcDQN, strategy explains a large amount of the variance (hcDRQN: 27%; animal: 37%).

In summary, our neural dynamics analysis suggests that the hippocampus is indeed involved in task strategy, temporal integration, and reward-based decision, in a way that is best matched by hcDRQN RL agents trained in partially observable environments.

## Hippocampal RL agents with recurrence better capture animal behaviour

Subsequently, we wondered whether RL agents could also capture animal behaviour during the task. To juxtapose the actions of RL agents against those of animals, we examined the trajectories observed during maze navigation post-learning. In order to facilitate a direct comparison with agent trajectories, we discretised the animal trajectories into a 9 × 9 grid. The behavioural trajectories show that hcDRQN better captures animal behaviour in terms of time spent at the starting point, decision, and the terminal state (Fig. 4a). Interestingly, we find that both models tend to spend more time in the decision state compared to the initial state, which shows that agents take some time before committing to a final decision. This observation is related to vicarious trial-and-error and increased dwell times often found in rodents when approaching decision states[46]. Note that agents can stay in a given cell because they can continue rotating left or right before deciding to move forward. This behaviour is highlighted by cells with darker red (Fig. 4a). On the other hand, as expected, the hcDQN agent fails to discriminate between the two allocentric subtasks and instead learns only one policy (allocentric south; Fig. 4a, middle). Next, to quantify the time spent on each state, we calculated the ratio between the time spent in individual states and the final state. This state-to-end ratio shows that hcDRQN also better approximates animal behaviour at this finer level and for allocentric strategies in particular (Fig. 4b, c). Next, to study whether the better fit of hcDRQN to animal behaviour is specific to a subtask, we made this analysis across all possible subtasks (Fig. 4d), and our results show that hcDRQN outperforms hcDQN. When analysing RL agents trained with full observability, we observed that hcDQN agents are still unable to provide a good match of behavioural data (Fig. 4c, d; Figs. S12, S13). However, hcDRQN trained in

fully observable environments provides a slightly better match with behaviour (Fig. 4c, d), demonstrating that considering neural dynamics (as above) and task generalisation (see below), in addition to behaviour, is important when comparing RL agents to animals.

Overall, hcDRQNs provide a better match to animal behaviour, further supporting the role of hippocampal recurrence.

## Recurrent agents capture more spatial information

As commonly observed in ANNs trained in spatial environments[17–21,25,30], we expected our models to also develop spatial encoding. To test the importance of recurrence in spatial coding, we calculated spatial coding metrics (see Methods)[21,47,48]. Our results reveal that the hcDRQN model trained with partial observability develops more compact and spatially localised representations when compared to all the other models (Fig. 5a; Fig. S14). Moreover, the hcDRQN model achieves significantly higher spatial information and lower sparsity, indicating more selective and efficient spatial coding, compared to the other models (Fig. 5b; Fig. S15, Table S3). These distributions align with values reported in dorsal CA1[49].

We also performed PCA on the population activity to visualise how the models encodes spatial information (Fig. S16). The hcDRQN model exhibits smooth, continuous trajectories with clear spatial structure by position, reward, and step number compared to hcDQN. Importantly, hcDRQN exhibits shared latent structure across north and south starting locations while models trained with full observability show abrupt clustering, suggesting over-reliance on direct sensory input rather than internal shared memory dynamics.

## Agent's behaviour predicts state-dependent action values

Because the recurrent RL agent is able to solve all tasks we expected this agent to yield state-action value predictions with higher and more certain Q-values when compared to the non-recurrent agent. To examine this in more detail we analysed the action-values for each state. First, we highlight the sequence of actions that makes hcDQN take the wrong arm and the correct policy learned by hcDRQN for both allocentric tasks (Fig. 6a). As expected, on average, hcDRQN has higher Q-values than hcDQN, which reflects the fact that it learns all tasks (Fig. 6b, bottom). Next, we studied action selection certainty by calculating the relative Q-value variance across all possible Q-values for a given state (Fig. 6b, top). This analysis shows that hcDRQN starts with lower action certainty but that it gradually increases over states until the terminal state. This reflects the effect of appropriate cue integration towards a decision, which can also be observed in models trained with full observability (Fig. S17 and S18). In contrast, hcDQN becomes more certain (i.e., lower variance) after the initial state, becoming again more certain once the decision is made. This result is explained by the relative variance obtained in allocentric tasks (Fig. 6b, right), which is a consequence of the hcDQN's inability to learn allocentric tasks (Fig. 2d).

The observation that the hcDQN becomes overconfident in the allocentric task, which it fails to perform, is reminiscent of the Dunning-Kruger effect. This phenomenon suggests that expert individuals often exhibit less certainty than those who are less knowledgeable or skilled[50].

## Network recurrence enables generalisation to stochastic environments

Until now, we have trained RL agents in environments in which cues are always present. However, both biological and artificial recurrent neural networks can also integrate evidence over time in probabilistic tasks[51–53]. To test the effect of stochasticity on the different agents, we created environments in which cues randomly appear and disappear (Fig. 7). We study two scenarios: (i) incremental random cue removal during inference (i.e., after learning) and (ii) effect of random cue

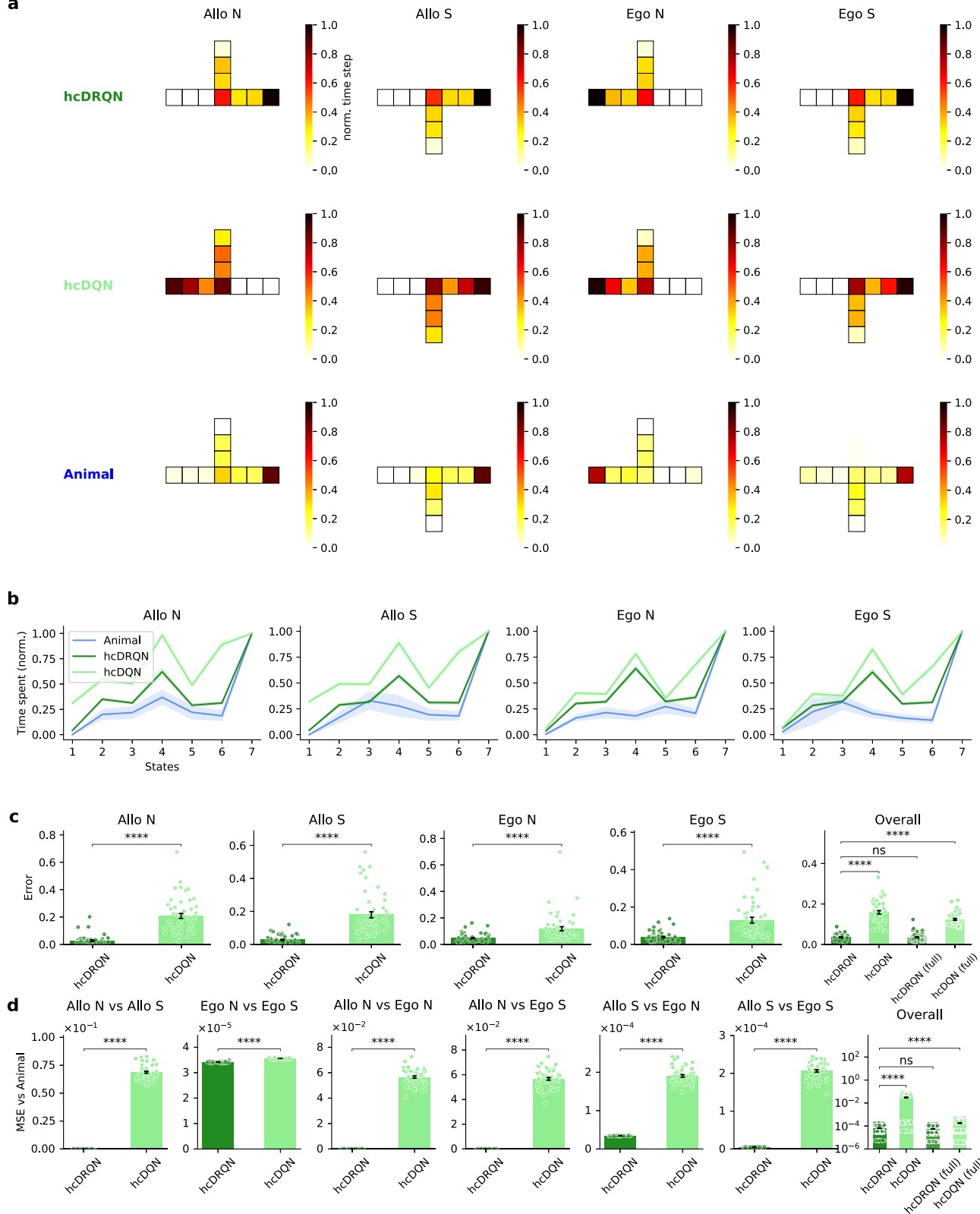

**Fig. 4 | Hippocampal RL agents with recurrence capture animal behaviour. a** Time spent on each maze state across the four strategies for rats, hcDRQN, and hcDQN. Animals spend more time at the decision and rewarded terminal points. hcDRQN better captures animal behaviour and hcDQN fails to solve the allocentric north task (cf. Fig. 2). **b** Time spent on each state normalised to the time spent on the final (terminal) state in models and animal. The 2D maze is represented as a series of discrete states along the optimal path from the starting point to the terminal point. Specifically, each of the 7 states corresponds to a unique position along this optimal 1D trajectory within the maze. **c** Error of time spent across states between behaviour predicted by the models and animal behaviour. **d** Model-animal behavioural errors (as in **c**) for each possible task-pairs. hcDRQN shows overall closer match to animal behaviour when compared to hcDQN. Data in (**b**, **c**, **d**) are presented as mean values ± SEM ($n = 50$). Exact $p$-values are reported in Tables S4-S5. ****: $p < 0.0001$, ns indicates no significant (independent two-sample, two-sided t-tests across models). Source data are provided as a Source Data file.

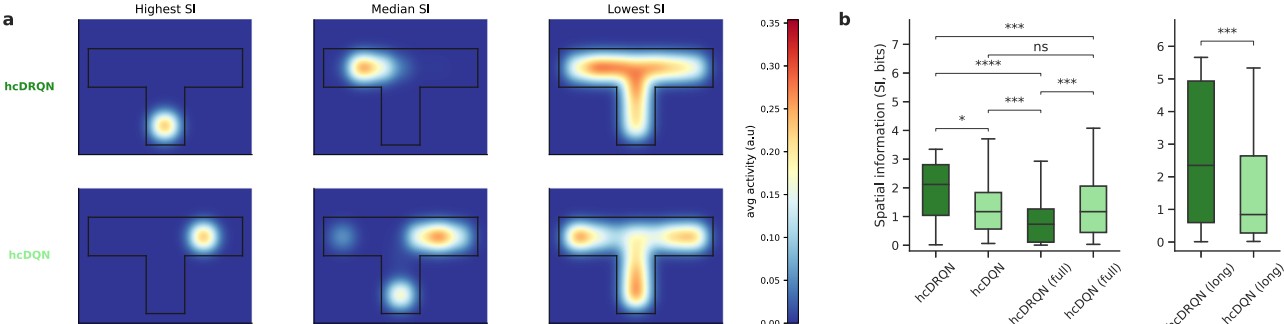

**Fig. 5 | Recurrent agents achieve higher spatial information in partially observable environments. a** Example place fields for hcDRQN and hcDQN neurons with the highest, median and lowest spatial information. **b** Spatial information (SI) distributions for all CA1 model neurons across hcDRQN and hcDQN for both the baseline environment (left) and the environment with a longer maze (right). Box plots show median (center line), interquartile range (box = Q1-Q3), and whiskers extending to the most extreme data within 1.5 × IQR from the hinges; points beyond the whiskers are considered outliers and are not displayed. Thus, the plotted min/ max are the whisker endpoints, not the global extrema. Two-sided independent-samples t-tests on per-neuron SI values ($n = 200$). $p_{hcDRQN\ vs\ hcDQN} = 1.95 \times 10^{-2}$, $p_{hcDQN\ vs\ hcDRQN(full)} = 2.38 \times 10^{-4}$, $p_{hcDRQN(full)\ vs\ hcDQN(full)} = 5.16 \times 10^{-4}$, $p_{hcDRQN\ vs\ hcDRQN(full)} = 1.22 \times 10^{-8}$, $p_{hcDRQN\ vs\ hcDQN(full)} = 4.56 \times 10^{-4}$, $p_{hcDQN\ vs\ hcDQN(full)} = 3.56 \times 10^{-1}$. $p_{hcDRQN(long)\ vs\ hcDQN(long)} = 6.45 \times 10^{-4}$. *: $p < 0.05$, **: $p < 0.01$, ***: $p < 0.001$, ****: $p < 0.0001$, ns indicates no significant. Source data are provided as a Source Data file.

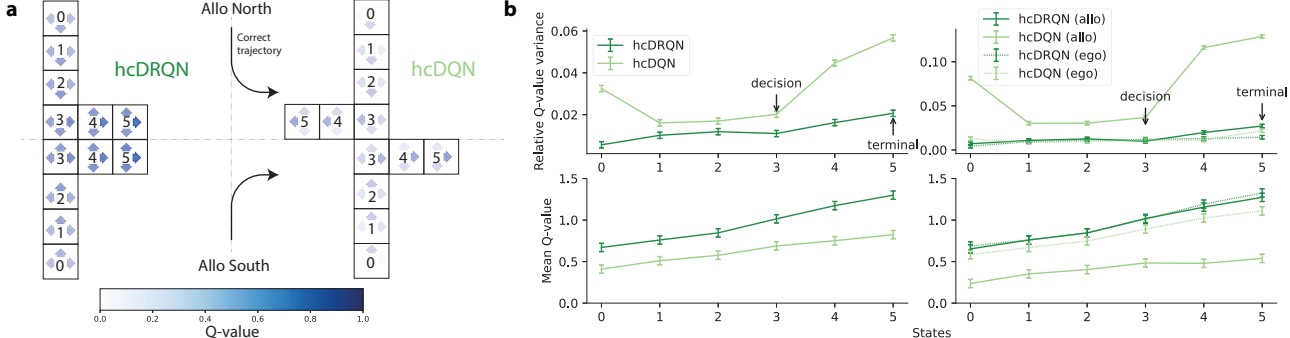

**Fig. 6 | Agent's behaviour predicts state-dependent action values. a** Schematic illustrating state-action policy for hcDRQN and hcDQN for allocentric subtask. It highlights that hcDRQN solves the task and that it is more uncertain about which action (i.e., different actions have similar values) until the decision point. Each state is represented by 3 coloured arrows corresponding to the state Q-values where darker colour means higher value. Note that the agent is capable of rotating left or right at any position, including the start of the maze. Rotating changes the agent's orientation but not its position, which is why there are non-zero Q-values associated with the left and right actions even at the starting location. **b** Top: Relative Q-value variance, given by the variance over Q-values for each state divided by the mean Q-value for each state. It shows that hcDQN decreases its relative variance as it gets closer to the decision state (see arrows) and that it has higher overall relative variance compared to hcDRQN. Bottom: Average Q-values over environmental states. Data are presented as mean values ± SEM over 5 different initial conditions. Source data are provided as a Source Data file.

removal on learning. During inference, the performance of hcDRQN gradually declines as cue availability decreases, reflecting its dependency on sensory cue integration (Fig. 7a). In contrast, hcDQN shows no change in performance across different cue probabilities (Fig. 7b), indicating that it does not rely on cue input and instead follows a fixed policy. Notably, hcDRQN maintains performance above 60% even at 90% cue removal, demonstrating robustness to reduced sensory input. When comparing the hcDRQN trajectories to the model without cue removal, the agent switches to the default 'turn right' policy when little or no cues are present. In addition, the model spends more time in the start and middle corridors because it must observe cues before deciding which arm to turn onto. Next, we trained the hcDRQN in a stochastic environment, where cues are randomly removed (Fig. 7c). This analysis helped to confirm that, as expected, RNN-based models can learn to integrate in stochastic environments. Indeed, we observed that training with stochastic cue removal led to more robust performance during testing (~10% improvement). Interestingly, when analysing trajectory behaviour, our model shows that agents trained with probabilistic cues spend more time at the starting location to integrate sensory evidence for longer before committing to a decision (Fig. 7d).

This result is in line with experimental observations showing that rats integrate sensory evidence over time and take longer to respond when this evidence is more ambiguous[54,55].

Taken together, these results suggest that recurrence plays a critical role in learning to navigate stochastic environments, consistent with the broader importance of recurrent dynamics for sensory integration in the brain[51,53,54,56].

### hcDRQN agent generalises to different task conditions
Finally, we tested whether the RL agents considered here can generalise to different task conditions not experienced during training (Fig. 8a). First, we tested different lengths of the initial maze corridor (Fig. 8b). This allowed us to test whether the RL agents memorise the tasks or learn to integrate the cue information and maintain it in memory to trigger the right action at the decision point. Our results show that hcDRQN is robust to different maze lengths in both allocentric and egocentric trials. This demonstrates that indeed the hcDRQN has successfully learned to integrate cue information, which is then maintained in its recurrent memory for action selection when required. The hcDQN is able to generalise within the egocentric task it

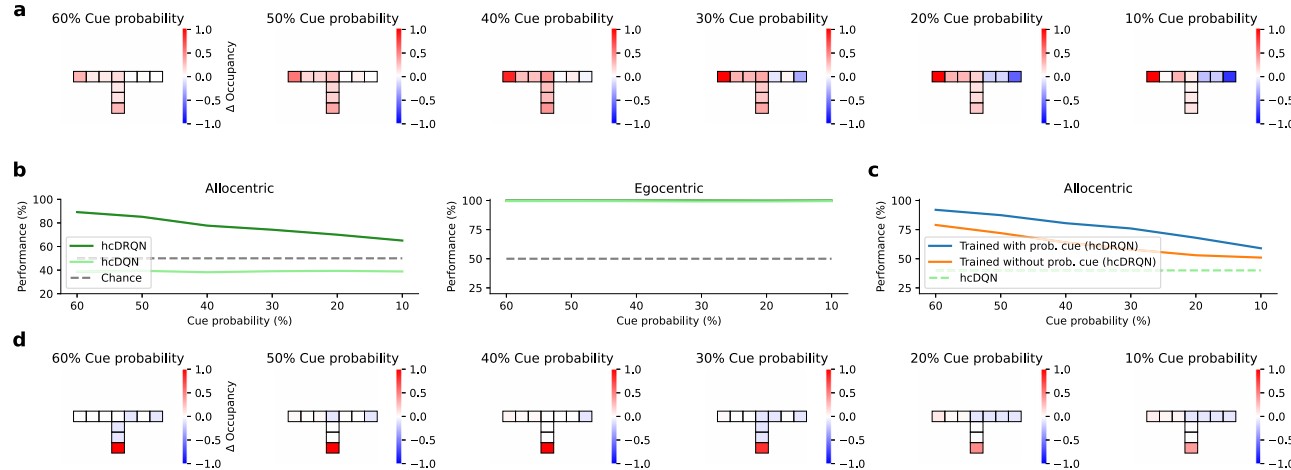

**Fig. 7 | Recurrence enables better generalisability to stochastic environments. a** Change in state-occupancy (with probabilistic cues - without probabilistic cues) for the hcDRQN agent across different degrees of cue removal. **b** hcDRQN outperforms hcDQN across different degrees of probabilistic cues. **c** Performance of hcDRQN agents trained with probabilistic cues compared to without. The hcDRQN was trained 80% cue probability. **d** Change in state-occupancy of hcDRQN agents trained with probabilistic cues. Data are presented as mean values ± SEM over 5 different initial conditions. Source data are provided as a Source Data file.

has learned, but this is primarily due to its ability to map specific states directly to actions, rather than through a model of the task or memory of previous states. This limitation is evident in its failure to generalise to trials involving distractors, as shown in the distractor condition below (Fig. 8).

Next, we tested the models on the same environment it was trained on, but removing a set of cues at a time. Note that this is a complete removal of cues, rather than stochastic cues as in the previous section. When retaining all cues the performance obtained by all models is in line with the training performance, with hcDRQN being the best model and the only one doing better than chance in the allocentric tasks (Fig. 8d, all cues). This demonstrates that the models were able to remember all the tasks on which they were trained. To test for generalisation, we gradually reduced the number of cues available in the environment. Our results show that hcDRQN is the only model that can handle half of the cues being removed (Figs. 8d and S19 for the same test with a longer maze). Interestingly, cue removal is more detrimental to allocentric navigation than egocentric navigation. This result is in line with experimental observations, in which allocentric but not egocentric task performance is impaired upon cue removal[57] (Fig. 8c). In contrast, models trained with full observability cannot generalise, as they rely on the presence of specific cues (Fig. 8g and h). Furthermore, the performance of both allocentric and egocentric tasks under full observability declines proportionally with the removal of cues. In both cases, the models fail to complete the tasks when no cues are present. This is in contrast to the partial observability version, where hcDRQN shows some degree of robustness to cue removal (cf. Fig. 8d).

Finally, we repeated the original task on the T-maze, adding a distractor cue and adding (Gaussian) white noise to the cues. When a distractor cue is introduced, hcDQN's performance drops to chance level, whereas hcDRQN achieves a success rate of approximately 70% (Fig. 8e). To test the robustness of both models to noise we tested a range of noise levels (Fig. 8f). The hcDRQN model can handle a relatively large degree of cue noise without any changes in the overall performance, while hcDQN is more unstable and shows a faster decrease in performance as the noise is increased.

Taken together, our generalisation tests demonstrate that hcDRQN generalises better to different and realistic task conditions, in line with animal behaviour.

## Discussion

Naturalistic behaviour almost always relies on navigating environments with limited visibility. Here, we have shown that recurrent hippocampal networks play a pivotal role in such environmental setups. Our investigation began by training RL agents to perform ego-allocentric tasks within partially observable environments. Remarkably, agents equipped with recurrent hippocampal circuitry successfully mastered these tasks, mirroring real-world animal behaviour. Additionally, our models predicted reward, strategy, and temporal neuronal representations, which we validated using tetrode hippocampal neuronal recordings. Furthermore, hippocampal-like RL agents predicted state-specific trajectories which resemble experimentally observed animal behaviour. In stark contrast, agents trained in fully observable environments failed to replicate the experimental data. Importantly, hippocampal-like RL agents with recurrence demonstrated enhanced generalisation capabilities across novel task conditions.

Motivated by the challenging conditions that animals often face in the wild, we have focused on a task setup with partial observability. This is also supported by the lack of visual acuity in rodents[58,59]. In addition, when our models were trained with full observability, they could not generalise (Fig. 8h), which further suggests that partial observability provides a better model of animal behaviour. hcDRQN performs particularly well in partial environments, in line with previous research in artificial neural networks[60]. Partially observable environments represent a more realistic setup, which, together with our results, suggests that hippocampal recurrence may have evolved to support the ability to navigate in these conditions.

Our model shows that hippocampal recurrence is needed to solve all the tasks we tested due to its ability to remember the relevant sensory cues. This is in line with experimental results showing that the CA3 region is involved in maintaining working memory representations in delayed-to-match sample tasks[9–11]. Moreover, using low dimensionality analysis we have shown that hippocampal neuronal dynamics encode a difference between correct and incorrect trials. This could be a reflection of animals taking different trajectories on incorrect and correct trials or reward value transmitted from downstream brain areas as predicted by temporal different reinforcement learning algorithms[61,62]. Furthermore, our work suggests that CA3 recurrence or other forms of hippocampal recurrence enable hippocampal networks to develop representations that are context-invariant, which can then be used by downstream decoders for action selection and reinforcement learning.

While here we have focused on a role of a CA3-like recurrence in guiding learning to navigate partially observable environments, we acknowledge that CA3 is not the only source of recurrence within the

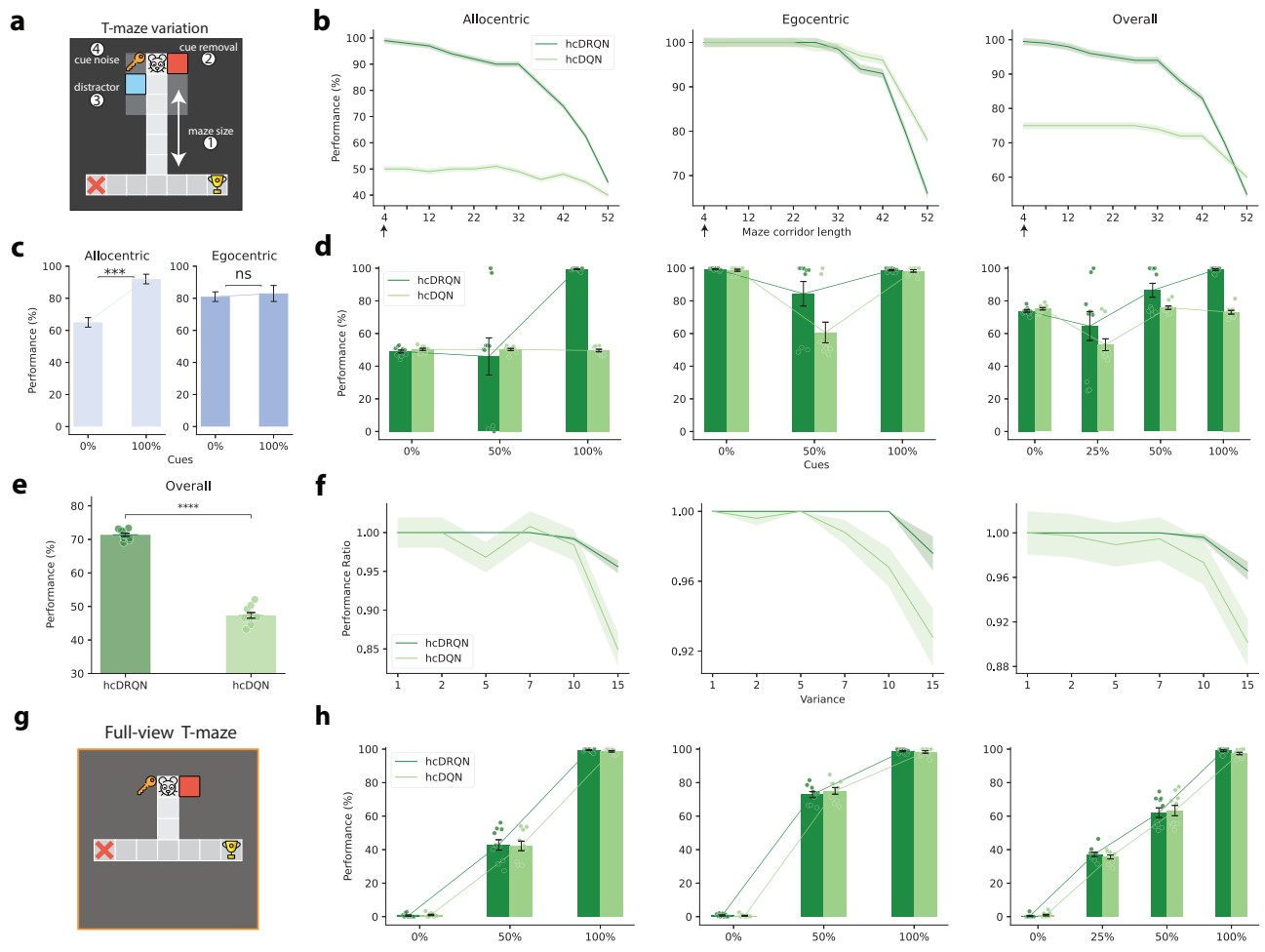

**Fig. 8 | hcDRQN shows better generalisation to maze length, cue removal, distractors and sensory noise. a** T-maze setup for increased length of the middle corridor, cue removal, distractor cue and random noise. **b**, Performance decrease over gradual increase of the maze length shows that hcDRQN can handle middle maze length being 32 steps. Black arrows on the X-axis represent maze corridor length of 4 steps utilised during training. **c,d**, Animal and model performance when cues are removed from the environment. Both hcDRQN (light green) and animal (blue) allocentric navigation are highly dependent on cues while egocentric is not affected by cue removal. On the other hand, hcDQN fails to solve both allocentric tasks. Data represent mean ± SEM ($n = 12$, $p_{allo} = 0.0004$, $p_{ego} = 0.53$, two-sided Wilcoxon rank-sum tests). ***: $p < 0.001$, ns indicates no significant. **e** When adding a distractor cue, hcDQN drops to chance level while hcDRQN can still solve most of the tasks. Significance stars indicate two-sided independent-samples t-tests across

models ($p = 2.01 \times 10^{-15}$). ****: $p < 0.0001$. **f**, When adding white Gaussian noise to the cues hcDRQN is more stable and robust when compared to hcDQN. **g** T-maze setup with full observability. This setup was used to evaluate generalisation capabilities following cue removal (**h**). **h** Compared to models with partial observability, all models with full observability show a drop in performance when cues are removed. Data are presented as mean values ± SEM over 10 different initial conditions for (**d**, **e**, **h**) and over 5 different initial conditions for (**b**, **f**). Source data are provided as a Source Data file. Icons used in panels a,g are released by OpenMOJI under a Creative Commons Attribution ShareAlike license 4.0. Panel **c** was adapted from H Malagon-Vina, S Ciocchi, J Passecker, G Dorffner, T Klausberger. Fluid network dynamics in the prefrontal cortex during multiple strategy switching. Nature Comms 309 (2018); reprinted with permission from Springer Nature under a Creative Commons Attribution 4.0 International License.

hippocampal formation capable of supporting such computations. Notably, prior work by Schapiro and others[12–14] has demonstrated that "big-loop" recurrence—mediated by bidirectional connections between entorhinal cortex (EC) and CA1 via the monosynaptic pathway—can also support statistical learning and temporal integration. This circuit provides an alternative mechanism by which the hippocampus might associate past sensory cues with future states, especially in scenarios requiring the accumulation of information across episodes. Future work should consider the multiple hippocampal loops, which can then be evaluated in light of the framework that we introduce here, but also that of others[12–14].

Our study generates experimentally testable predictions on the role of recurrence in facilitating navigation in environments with limited visibility. In particular, our work predicts that perturbing hippocampal recurrence (e.g., CA3) should impair the ability of animals to generalise to new and stochastic task conditions. Furthermore, our

findings suggest that plasticity in such recurrent synapses is important in supporting these functions (Fig. 2d).

Because our focus is on contrasting models with neuroscientific observations, we have used the same block-wise training of ego- and allocentric tasks as employed experimentally. In this setup, our work predicts that hippocampal recurrence is particularly important to learn allocentric tasks (Fig. 2d). Generalisation tests, show that our models are more robust to the removal of cues in egocentric tasks when compared to allocentric tasks (Fig. 8c), consistent with experimental observations[57] and suggesting that the agents have learned the macrostructure of the task. However, the tasks that we have used here represent only a small subset of all the possible challenges that animals are faced with. This means that our model-animal comparison is relatively unfair, as animals have to deal with much more than solving these two tasks. Relatedly, we have assumed the existence of action-value output networks that perform clean allo-egocentric task-

switching. While such computations are believed to be executed by striatal networks[63], the precise mechanisms remain elusive. Finally, we have used artificial algorithms for credit assignment in hippocampal networks. All these elements remain to be explored in future work.

Our results show that in the multi-task block-training setting we tested, a model with a recurrent layer (hcDRQN) without experience replay outperforms alternative methods that were specifically designed for continual learning, consistent with recent machine learning findings[64]. Given that hcDRQN is a systems-level approximation of the hippocampal system, it suggests that the brain relies on a combination of recurrent neural networks to continually adapt to new situations, at least in navigational tasks. It remains to be tested how general these principles are across other areas of the brain.

In our work, we have either assumed task-specific output heads (Figs. 2–4) or task ID information (Fig. S2) to enable agents to solve multiple tasks. However, other solutions exist that are more biologically plausible. One possibility is that the brain mitigates interference between tasks through orthogonalisation of neural representations[65–67]. Another hypothesis is that the brain performs latent-state inference, using accumulated evidence (e.g., reward history, environmental cues) to infer which task is currently active[68,69]. Although our current models do not implement such mechanisms, integrating a latent-state inference component or learning orthogonal task representations would be an interesting direction for future work, and could help bring our models closer to the biological solutions employed by rodents in blocked task settings.

Classical hippocampal models suggest that recurrence enables pattern completion[70,71], while more recent computational models propose that the hippocampus creates a predictive map of the environment through successor representations[25]. Our research aligns with this predictive view of the hippocampus, because in our model the hippocampus learns to predict the correct trajectory to take given sensory input, which is then reflected in simulated and experimental neuronal dynamics (Fig. 3a). Furthermore, our findings underscore the essential role of recurrence in constructing the hippocampal predictive map, consistent with the predictive view of hippocampal function[17–21,25,30]. Our work complements this view by showing that training RL agents under partial observability is important to capture experimental observations. In another set of studies recurrent neural networks (RNNs) have been trained to support spatial navigation. They have shown that RNNs develop spatial representations similar to experimental findings[17–19]. For instance,[18] demonstrated grid-like spatial response patterns, border cells, and band-like cells in trained RNNs. Similarly,[17] revealed grid cell-like representations when training deep recurrent reinforcement learning networks for 3D navigation.[19] trained a similar system, yielding neurons with spatial properties akin to those in refs. 17,18. While these studies emphasize the importance of memory (see also ref. 20) and recurrence in hippocampal networks, they do not assess their function in partial observability and its relationship to experimental observations in goal-driven tasks, which is our focus. In future work, it would be of interest to combine our RL-based approach with self-supervised approaches, as these probably provide complementary functions[17–19,21,30].

In comparison, other models have explicitly considered reward-based navigational tasks. For example, work by ref. 27 utilises an actor-critic network and goal-independent representations for solving water-maze tasks, which was later extended hierarchically by ref. 29. On the other hand,[31] combined external vision, path-integration and goal-driven reinforcement learning in a model evaluated on a physical robot. Whilst these models do connect various components to distinct brain structures, they have largely overlooked the significance of hippocampal recurrence and its implications in real-world scenarios, including partial observability. Our research highlights the critical nature of these features in accurately modelling both behaviour and neuronal dynamics.

A limited number of models in neuroscience have considered environments with partial observability[72,73]. A notable exception are Clone Structured Cognitive Graphs that create latent state-space maps for aliasing (i.e., partial observability) while capturing cognitive map phenomena[73,74]. Our work adds to this body of literature supporting the importance of partial observability in explaining experimental observations.

Overall, our work suggests that hippocampal networks play a critical role in the ability of animals to continuously adapt to the environment under realistic conditions and with good generalisation properties.

## Methods
We begin by outlining the deep reinforcement learning approaches employed in this study, followed by an explanation of the methods utilised for the analysis of neural data.

### Reinforcement learning models
We developed a deep reinforcement learning model consistent with the hippocampal architecture. To train the models we designed a custom-built 2D gym-minigrid maze[33], mimicking the T-maze environment (Fig. 1a) in line with common experimental setups, which allows us to compare our models with the behavioural and neural data[5]. In order to capture cues commonly placed on external walls in experimental setups, we placed four cues in the environment in both allocentric and egocentric trials, in line with ref. 5. At the beginning of any given trial, the agent was placed at the start of the north or south arms of the maze following the same setup of the animal experiments, and we closed access to the opposite arm, thus converting the maze into a T-maze. We considered two terminal states: one rewarded and one unrewarded. After reaching a terminal state (rewarded/unrewarded) we allowed the agent to continue exploring for three extra time steps, which allowed us to model the animal behaviour right after reward consumption. During model training, we extracted neural activities that we used to contrast model and animal neural data. Note that direct sensory information about the terminal states was not given as input to the agent.

**Deep RL agents.** Reinforcement learning (RL) models an agent that observes the environment and takes an action $a$. This action transitions the agent into a new state $s$ of the environment, which might give back a reward $r$ according to the utility of the action selected. This can be formally defined by a Markov Decision Process as tuple of $\langle \mathcal{S}, \mathcal{A}, P, \mathcal{R}, \gamma \rangle$ where $\mathcal{S}$ is the set of all the states, $\mathcal{A}$ is the action set, $\mathcal{P}$ the transition matrix $P(s'|s, a)$ from current state $s$ to the next state $s'$ when taking action $a$. The objective is to maximise the expected total rewards, called return $G_t$ defined as $G_t = \sum_{k=t}^{T} \gamma^{k-t} R_{k+1}$ where $t$ is the current time step, $R_{t+1}$ is the reward obtained at time $t + 1$, $\gamma$ is a discount factor such that $0 \leq \gamma < 1$ and $T$ is the time at which the episode terminates.

For the hippocampal RL models, we build on the standard deep reinforcement learning models. In particular, we use Deep Q-networks (DQN)[34], in which states $s$ are provided as input to an artificial neural network that are then mapped onto value-action pairs $Q(s, a)$. The network is trained using state-outcome transition tuples, $(s, a, s', r)$, where $s$ is the current state, $a$ is the action, $r$ denotes reward outcome and $s'$ the next state. The error function used to train the hippocampal network follows a Q-update function as $E_i$ at step $i$:

$$E_i(\theta_i) = E_{s,a,s',r \sim D}[r + \gamma \max_{a'} Q(s', a'; \theta_i^-) - Q(s, a; \theta_i)]^2 \tag{1}$$

where $\theta$ denotes the network weights, $\gamma$ is the discount factor and $D$ is the dataset of past trajectories. As done by standard DQNs we use the concept of target network ($\theta^-$). The target network is a frozen version of the primary neural network, which helps to stabilise reinforcement

learning. It is updated at regular intervals to avoid inflated estimations of the Q-values. The hyperparameter target update counter controls the frequency of updates of the target Q-network. Equation (1) calculates expectation of the squared error between the target Q-value and the predicted Q-value over a batch of state-action experience tuples. We minimise this expected error by adjusting the parameters $\theta_i$ of the network using standard gradient descent and Adam optimiser (see parameters in Table S1).

To create a model that more closely captures the hippocampal circuitry, we consider a three-layered structure with the input layer modelling entorhinal input to Dentate Gyrus (DG), the first layer represents CA3, a second layer represents CA1, and the output layer encodes the value of a given action-state pair, $Q(a, s)$ (Fig. 2b). The DG input originates from the entorhinal cortex (EC) which is believed to provide the hippocampus with a spatial map of the environment[23]. In our model, the EC spatial map is approximated by the 2D top-down spatial map provided by the minigrid environment (Fig. 1c). The output layer encodes state-action Q values, which abstracts out hippocampal-to-striatum functional connectivity[35]. A Q-value output action head was not sufficient to solve all tasks that we consider, even in the presence of replay mechanisms (Fig. S3a). Therefore, to ensure that the network could solve the two tasks (ego and allocentric) we use task-specific heads (i.e., two heads) at the output (see Methods), for all the models we consider. In biological networks this could be implemented through task-specific contextual signals[75]. Indeed, it is also possible to train agents with only one head, provided that a task-specific contextual signal is provided as input to the model (Fig. S2). Note that it is only at the output level that we assume task-separation (i.e., one output head for allocentric and another for egocentric tasks). This means that the representations developed by the hippocampus have to be generalisable across the different tasks. Given that we can capture key aspects of experimental data with this setup, this implies that task-specific Q-action values are encoded outside the hippocampus.

All our models have a three-layered (hippocampal-like) structure in which the input layer is of shape ($RxC$) where $R$ is the number of rows in the input grid, $C$ is the number of columns. The output layer has $Nx1$ shape ($N = 3$) denoting the Q-values for the 3 actions that the agent can take in the environment (left, right, forward). We use a standard discount factor ($\gamma = 0.9$), no replay memory (i.e., replay memory buffer size = 1, but see Fig. S3 for different variants) and a batch size of 1 during training. Adam is used as the optimiser with a learning rate $\alpha$ of 0.001. We use a CA3 layer size of 50 for all models considered. In the case of hcDRQN these units are Gated Recurrent Units. A Gated Recurrent Unit (GRU)[40,41] network is a specially designed recurrent neural network that can capture long temporal dependencies in data. GRUs achieve this through a gating mechanism, which regulates the flow of information. This mechanism includes an update gate, controlling the extent to which a GRU unit keeps the previous state, and a reset gate, deciding how much past information to forget. The learning rate and $\epsilon$ (0.3 − 0.05) for $\epsilon$-greedy, which controls the trade-off between exploration (choosing new actions) and exploitation (using known actions) have been selected using a grid-search. All the hyper-parameters are given in Table S2.

**Partial observability.** In our grid-based environment, models operate under conditions of limited environmental observability, mirroring the real-world challenges faced by navigating animals. Moreover, in reality, animals rarely possess complete access to all pertinent sensory data during navigation. For instance, they may initially focus on cue information but then shift their attention to executing motor commands to reach their destination. In experimental neuroscience, while cues are typically positioned along the outer walls of a room[5], animals do not continuously fixate on these cues. Additionally, maze setups often involve the incorporation of walls of varying heights, further restricting the visual input available to the animals.

To substantiate the importance of CA3 in navigation and its ability to better align with experimental findings in partially observable environments, we compare our models against those trained with full visibility. This comparison underscores the significance of CA3 and its capacity to more accurately capture experimental outcomes when navigating in environments with limited sensory input.

**Training details.** The training phase consisted of a block of allocentric and egocentric trials. Specifically, each block contains 25 trials, and there were a total of 4 blocks (allocentric north/south, egocentric north/south). To model egocentric trials the reward location changed between north and south, but for allocentric trials it was kept constant. The agents were first exposed to blocks of allocentric trials in the north direction, which were then alternated with blocks of allocentric trials in the south direction. This alternating pattern was repeated four times before switching to egocentric trials. The same north/south combination was maintained throughout the duration of the egocentric trials. In total, the training consisted of 10,000 individual trials (200 blocks each with 25 trials for both ego and allocentric tasks).

All model parameters (weights and biases) were updated using BPTT within the trial and once the model reached the terminal state (i.e., there is no BPTT across trials and batch-size=1). Further, BPTT is applied at every two steps backwards in time until the first step is reached, as is commonly done in deep RL[76]. The maximum number of steps that BPTT can go back in time is 128 steps, which is also the max number of steps during inference. hcDQN is trained in a similar fashion but we do not backpropagate the gradients in time towards the first step. During model testing, such as during the generalisation tests, we turn off learning/plasticity.

A two-head setup is utilised, where the final layer outputs two Q-values: Q-value-allo and Q-value-ego. The state input to the models are 2D matrices where cues, walls and no-walls were encoded as scalar values. For the partial view the observation size was 9 (3 × 3) while for the full view was 81 (9 × 9).

**Generalisation tests.** We performed four types of generalisation tests (Fig. 8):

1. *Longer maze*: In this test, the length of the starting corridor was increased while keeping the length of the two terminal arms constant.
2. *Cue removal*: Different combinations of cue removal were performed, ranging from removing all cues to removing none. The cues were removed at the beginning of the trial, meaning that the agent had no access to the cue at any point during the task. This is in contrast to the experiments on probabilistic cue removal (Fig. 7), where the agent still had access to these cues with a given probability.
3. *Distractor*: Another cue (represented by a scalar value) was added just next to (below) the existing cues.
4. *Random noise*: We added normally distributed noise, $\mathcal{N}(\mu, \sigma^2)$ with $\mu = 0$ and $\sigma^2$ in the range of (1,15).

**Continual learning algorithms.** Continual Learning, also known as Incremental or Life-long Learning, refers to the ability of a model to sequentially master multiple tasks without losing previously acquired knowledge, even when data from older tasks are no longer accessible during the training of new ones. We evaluated the performance of two state-of-the-art regularisation-based continual learning algorithms: Elastic Weight Consolidation (EWC)[42] and Synaptic Intelligence (SI)[43]. These artificial algorithms were tested against our recurrent and non-recurrent DQN architectures in identical task settings. EWC adds a penalty term to the loss function, derived from a Fisher Information Matrix (also called weight importance), which guides the learning such that information pertaining to previous tasks remains preserved. On the other hand, SI determines the importance of each neural network

weight throughout the learning process and uses this information to restrict weight updates, preserving prior task knowledge. Both importance measures, which are tunable configurations (hyperparameters), were selected through hyperparameter optimisation (the EWC weight importance was set to 800 and the SI weight importance to 30).

**Experiments with fully observable environments.** We repeated the main results with a fully observable environment with both hcDRQN and hcDQN models. Although both models can learn all tasks in terms of performance, their dPCA analysis does not show a clear separation of all the components. In hcDRQN, decision and strategy follow the same trends with activity dropping to zero right after the reward point. This is the opposite of the partial view hcDRQN and animal activity where strategy components keep the separation even after the reward point. Moreover, hcDRQN fails to capture the time component. The hcDQN model completely fails to separate the strategy north components. Overall, given that full view model fails to capture dPCA components we argue that the fully observable environment does not provide a good match of the hippocampal data. We run further tests to analyse the animal trajectories and the generalisation capabilities of these full view RL models. Although most of the trajectory maps show a close match between the animal and hcDRQN, there are situations in which the hcDQN seems to be a better match to animal data. The generalisation tests highlight the limits of the fully observable models, as cue are gradually removed performance drops drastically, emphasising the dependency of these models on the cues. Animal performance and partial view RL models show evidence that egocentric task do not rely on cues, however, the fully observable models remain highly dependent on the cues. Overall our results suggest that hcDRQN trained with partial observability provides an overall best match with animal behaviour and neuronal encodings.

**Computing details.** All experiments were conducted on the BluePebble supercomputer at Bristol; mostly on GPUs (GeForce RTX 2080 Ti) and some on CPUs (Intel(R) Xeon(R) Silver 4112 CPU @ 2.60GHz). We did not record the total computing time for the experimental results presented in this paper, but this can be estimated as follows. To train each model (one seed with all the task-specific trials) takes approx 1 hour and 30 min. For each of the models, we run 5 random seeds, resulting in approx 6 hours per model. When recording the activations, the total time is around 8 hours. Testing a single model for one seed takes approx 5 min. Overall total time it takes to run our models is 32 hours (8 x 4) for training with 5 seeds and 2 hours for the testing results with 5 seeds.

**Statistical analysis.** Due to the inherent variability of the starting conditions on the learning path of these models, we trained our models across 5 different randomly selected seeds. To assess the significance of all relevant figures, we conducted a two-sided paired t-test on the relative alterations across the various seeds. Significance levels are denoted as follows: * ($p < 0.05$), ** ($p < 0.01$), *** ($p < 0.001$), and **** ($p < 0.0001$).

**Spatial coding analysis.** To quantify the spatial coding of CA1 units, we computed three standard place-cell metrics[21,47,48] on discretised $x \times y$ occupancy maps. First, spatial information (SI) measures how much observing one unit activation reduces uncertainty about the agent's location. SI is expressed in bits and is high when a unit fires selectively in a small subset of bins. Second, the sparsity index captures how unevenly activity is distributed across space. A value near 0 indicates a sparse code (few bins active), whereas a value near 1 indicates dense, broadly tuned firing. Third, spatial coherence quantifies local smoothness by correlating each bin's firing rate with the mean of its eight neighbours; high coherence reflects well-formed, contiguous

**Table 1 | Nomenclature for spatial metrics**

| Symbol | Meaning |
|---|---|
| $i$ | Index over spatial bins |
| $p_i$ | Occupancy probability of bin $i$ |
| $r_i$ | Mean activity (firing rate) in bin $i$ |
| $\bar{r}$ | Overall mean activity, $\bar{r} = \sum_i p_i r_i$ |
| $\tilde{r}_i$ | Local mean activity (3 × 3 uniform filter) |
| $\epsilon$ | Small constant ($10^{-8}$) to avoid division by 0 |

place fields.

$$\text{SI} = \sum_i p_i \frac{r_i}{\bar{r}} \log_2\left(\frac{r_i}{\bar{r}}\right) \tag{2}$$

$$\text{Sparsity} = \frac{\left(\sum_i p_i r_i\right)^2}{\sum_i p_i r_i^2 + \epsilon} \in [0, 1] \tag{3}$$

$$\text{Coherence} = \text{corr}\left(r_i, \tilde{r}_i\right) \tag{4}$$

To compute these metrics for every neuron we (i) accumulated an occupancy map and an activity map across all episodes, (ii) divided the latter by the former to obtain the place field $r_i$, and (iii) evaluated the above formulas using only bins with non-zero occupancy (see Table 1 for details about the variables used). Metrics were averaged over neurons within each model and compared with two-tailed t-tests (Fig. S15, Table S3). This triplet of metrics jointly characterises information content, coding sparsity and field smoothness, providing a compact yet interpretable lens on spatial representations learned by the models.

### Neural and behavioural experimental data

**Neural data analysis using demixed PCA.** We used the neural activities of 612 hippocampal CA1 neurons from five behaving rats were recorded, which were obtained in the dorsal and ventral CA1 using multiple tetrodes (Fig. 3a; see full details in ref. 5). Spike sorting was used to assign spikes to different neurons (full details in ref. 5), which were then converted to firing rates of individual neurons using a bin size of 0.2s[5]. Neural activity was aligned to the crossing of a reward-sensor. We considered neural activity 2s before and 4s after crossing the reward sensor for dPCA analysis. The reward sensor was located 10 cm from the end of the rewarded area. Animals were trained on a T-maze task in which rats had to follow both allocentric and egocentric navigational rules to reach reward points. We performed demixed Principal Component Analysis (dPCA)[77] on the neuronal firing rates using 3 behavioural variables – trial decision, strategy and time (dPCA $\lambda = 2.919e^{-05}$ was found using grid-search as done by ref. 77). We used the dPCA code made available by the authors of[77] in https://github.com/machenslab/dPCA. For models, we aligned data to the timestep in which a terminal state was reached and converted timesteps to seconds by making one timestep correspond to 200ms. To contrast with animal data, we continued recording the neural activity for 4s after reaching the terminal. The animal experiments used in our work were approved by the appropriate ethics committee as in ref. 5.

To quantitatively assess the differences in neural dynamics between observed data and model, we calculated the mean squared error (MSE) for the principal components derived from dPCA of both the animal and model CA1 neural activities. The MSE represents the average of the squared differences between the observed dPCs ($Y_i$) and the dPCs predicted by the model ($\hat{Y}_i$), as given by MSE $= \frac{1}{n}\sum_{i=1}^{n}(Y_i - \hat{Y}_i)^2$. $Y$ and $\hat{Y}_i$ were normalised by their max and min values before computing this error metric.

**Behavioural data.** The behavioural data (i.e., animal task performance) consists of a total of 47067 trials recorded over multiple days (3 to 7) from a total of 5 animals[5]. However, as some animals only had a maximum of 800 continuous trials, we used a maximum of 800 continual trials per animal. The maze was cleaned with an odour-neutral solution every 10 trials to avoid odour-guided navigation as described in ref. 57.

## Reporting summary

Further information on research design is available in the Nature Portfolio Reporting Summary linked to this article.

## Data availability

The animal data used in this study are available at http://github.com/neuralml/hcRL. Source data are provided with this paper.

## Code availability

The code used for data analysis in this study is available at http://github.com/neuralml/hcRL. We used the PyTorch library for all reinforcement learning models. The code used in our simulations is also available at http://github.com/neuralml/hcRL, released in Zenodo (https://doi.org/10.5281/zenodo.17008842).

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

## Acknowledgements

We would like to thank the Neural & Machine Learning group, Kim Stachenfeld and Chris Summerfield for useful feedback. DP was funded by the EPSRC Centre for Doctoral Training in Future Autonomous and Robotic Systems (FARSCOPE) [EP/S021795/1], SM by the Wellcome Trust (205038/Z/16/Z) and BBSRC and RPC by the Medical Research Council (MR/X006107/1), BBSRC (BB/X013340/1) and a ERC-UKRI Frontier Research Guarantee Grant (EP/Y027841/1). HM was funded by a FWF grant (I 5458; part of the German Research Foundation Research Unit 5159) and a WWTF grant (CS18-039). SC was funded by a European Research Council starting grant 716761 and a Swiss National Science Foundation professorship grant 170654. This work made use of the HPC system Blue Pebble at the University of Bristol, UK. We would like to thank Dr Stewart for a donation that supported the purchase of GPU nodes embedded in the Blue Pebble HPC system.

## Author contributions

The author contributions are as follows: Conceptualization: D.P., S.M., and R.P.C. Methodology: D.P., S.M., H.M.V., S.C. and R.P.C. Investigation: D.P., S.M., M.V.D., H.M.V., S.C. and R.P.C. Visualization: D.P., S.M., H.M.V. and R.P.C. Supervision: R.P.C. Writing—original draft: D.P., S.M., and R.P.C. Writing—review and editing: D.P., H.M.V. and R.P.C.

## Competing interests

The authors declare no competing interests.
