## [Transparent Peer Review file · Nature Communications]

Hippocampus facilitates reinforcement learning under partial observability

Corresponding Author: Dr Rui Ponte Costa

Version 1:

Reviewer comments:

Reviewer #1

(Remarks to the Author)
Summary

The authors present a model of the role of the hippocampus in ego- and allocentric navigation tasks, and compare the model's behaviour and internal representations to that of rodents in a previously recorded dataset. The model consists of a deep Q-network (DQN), and the authors systematically vary aspects of the model's architecture and inputs, such as whether there is a recurrent layer, and whether the model has access to a full top-down view of the maze or a restricted partial view. The authors find that the recurrent model, but not the feedforward model, successfully learns to navigate in both allo- and egocentric conditions. Their analysis of the model's internal representations reveals that the recurrent model has similar low-dimensional representations to those found in the animal data, and the authors show that the recurrent structure of the network enables better generalisation to different task conditions.

Major comments

This paper addresses an interesting question, of how the brain manages to optimize navigation behaviour under partial observability: not all information needed for a decision is present in the current observation. This means there's a need for some kind of working memory, which is naturally modelled with a recurrent neural network. The modelling approach is impressive and interesting and the authors did a good job linking the model to neural data, but there are a few points in the paper where I think the phrasing is confusing, making it harder to understand what is really happening. Most importantly, it would be great to flesh out a bit more what neuroscientists should learn from these AI models.

The main issue I have is that comparing the two models (recurrent and nonrecurrent) in their fit to animal data seems somewhat unfair. The way that the RL environment is set up, it would be impossible for a feedforward network to learn the allocentric task from partial view, because not all information needed for a correct decision is present in the input at any given time. For a fair comparison, I would expect to see two models that can both solve the task to a similar degree that animals do. For example, between the recurrent model with partial view and a feedforward model with full view, such as the one shown in figure S7.

Secondly, the key difficulty about the behavioural task presented in this paper is inferring from the block which task (egocentric or allocentric navigation) must be completed at the current trial. The authors side-step this issue in their modelling efforts by assuming two different output heads for the allocentric and egocentric tasks (main paper), or by providing the task ID (fig S2). That is fine, but since that information is not available to the rodents I think it would be good to include a discussion on how the brain solves this blocked multi-task learning problem. For example, some authors have suggested that the brain employs methods for finding orthogonal representations to avoid catastrophic. Other work has suggested that the brain might switch between discrete representations corresponding to different tasks, using some Bayesian inference mechanism. It would be great if the authors could comment on how they think this would be achieved.

Finally, the presentation of the corridor length generalization in fig 7 is confusing. The authors write that the recurrent model "is very robust across a large range of lengths, whereas hcDQN defaults to chance level". However, hcDQN was already at chance level for the allocentric task, so this is not a test of (failed) generalisation but of failed learning. If anything, for the egocentric task that both models can learn, the results show that the feedforward model shows better generalization to

longer corridors. This makes intuitive sense to me, because it learns a feedforward mapping from the inputs to a policy: “if I’m in a corridor with nothing else, run forward” and “if I’m at a decision point turn right”. Such a policy will generalise, whereas the RNN dynamics, when trained on a single corridor length, will be specific to that corridor length.

Minor comments

- Line 58: at this point it’s unclear what a “hippocampal-like” network means.
- Line 75-76: “on a given trial rodents have to infer the task (ego versus allocentric) through trial and error.” This is confusing. I think the authors mean “in a given block [...]” since the rule inference depends on the block.
- Line 77-78: quibble with the phrasing here: “In such setups, and in most navigational task setups rats are unlikely to be continuously accessing the full environment, implying a degree of partial observability (28)”. In RL, full observability does not necessarily mean continuous access to the whole environment. It means the agent can access all information that is relevant to decision making.
- Supplementary Fig S1: I do not understand the argument that the change in head direction shows that the task is partially observable. It seems to me that the partial observability stems from the decision rule having to be inferred by the animal.
 - o Please annotate what is shown in these plots. The rightmost plot shows “difference”, between what and what? And what are the units?
- Fig 1B. According to the caption the significance stars here correspond to a “paired t-test”. I think this should be a one-sample t-test if what’s reported is the difference to chance level (50%)
- 117-118: “We did not use replay in our model because of the lack of evidence of replay from the hippocampus to itself”. Can the authors clarify what is meant by this? There is ample evidence for replay in the hippocampus. While many theoretical models assume that these are read out by cortex during consolidation, I am not aware of any more evidence for that than for read-out by the hippocampus itself. If this exists, please provide the references.
- 153: “trained in partially environments”, missing “observable”
- 157-170: the dPCA analysis needs more explanation. How was the given label of each component identified? Further, the authors write that “both hcDRQN and hcDQN exhibit a temporally decaying population dynamics [...] These features of the hcDRQN highlight its unique ability to integrate input information over time ...”. If both the hcDQN and the hcDRQN show this temporal decay, how does that highlight the hcDRQN’s unique ability?
- Fig 6. What’s the point of this figure? It makes sense in machine learning, but what can neuroscientists learn about this?

(Remarks on code availability)

Reviewer #2

(Remarks to the Author)

In this work, Pedomontie et al. demonstrate that including recurrence in a hippocampus-like neural network architecture allows agents to learn to associate past stimuli with future states, which is necessary for adaptive behavior in partially observable environments. This also gives rise to a series of behaviors and neural population responses that are similar to rats in a simple but classic task with (the plus maze with allocentric navigation and location-based cues) and without (the plus maze with egocentric navigation) task-relevant partial observability. Several extensions to the environment are also thoroughly explored. Overall, I enjoyed this paper and found the results to be largely sensible and important. I think the work will be interesting to many researchers in this space. I do however have a few points that I would like the authors to address in a revision.

First and most importantly, while I am convinced by the paper that recurrence is needed to associate past sensory cues with future states, I am less convinced that it is specifically CA3 recurrence that is required to accomplish this. This is because CA3 recurrence (via the trisynaptic pathway) is not the only type of recurrence in the hippocampal formation that can solve this sort of problem. “Big loop” recurrence via the monosynaptic pathway (e.g. bidirectional links between EC and CA1) has been shown to allow statistical learning, which yields another mechanism for the hippocampus to link past sensory cues with future states, and thus to solve the problem of partial observability (e.g. Schapiro et al., 2017). There is indeed no mention of this work in the current paper, and I think at minimum the authors should discuss this possibility. Claims throughout about CA3 recurrence being the primary hippocampal mechanism at play here need to be weakened unless the authors can show that a model with only CA3 recurrence captures this behavior better than a model with only a monosynaptic pathway. This seems particularly important because Anna Schapiro’s line of work demonstrates that the trisynaptic pathway actually fails to learn regularities across individual episodes in the presence of big-loop recurrence via the monosynaptic pathway. A large part of why this occurs is also because there are several other biologically-relevant traits of CA3 that the present work doesn’t explicitly account for, namely high inhibition and resulting sparse connections. Fully addressing this point would require accounting for these traits of CA3 as well as more explicit modeling of the EC, which I recognize is a substantial departure from the current model. But I think it is important to acknowledge that this abstraction is a major weakness of the current work relative to state-of-the-art models of the hippocampus. Ideally, this point should be addressed by comparing to an architecture that is more biologically-realistic and accounts for this possibility. If such modeling is not feasible, these weaknesses should be acknowledged with writing and framing overhauls throughout the paper.

I have several other line and figure-specific comments:

- Line 18: The way this sentence is written in the abstract makes it sound like the agents are performing some kind of dimensionality reduction themselves, but I believe that the authors mean they use dimensionality reduction techniques to

investigate this. This should be made more clear.

- Lines 40-42: The writing here makes it sound like all model-based approaches are successor representations (SR), which is of course not true, so that should also be made more clear.

- Lines 47-48: It is not clear from this brief mention why a hierarchical model accelerates learning, and why it is important to mention this.

Supplementary Figure S1: What is the difference plot? It was not clear to me what the y-axis represents here and it is not mentioned in the figure caption.

Figure 1A: From the results main text alone, it isn't clear if the sensory cue locations directly mimic those from the experimental setup. Maybe it'd help to represent the cue locations somehow in Fig 1A?

Figure 1B: Are these box plots over individual trials? Would it be more informative to see this broken out by individual subjects? Perhaps a violin plot in the background over individual trials (if this is indeed what the box plots represent) to give a sense of the distribution of trials with individual animal means plotted on top would be more informative?

Figure 2A: The caption mentions the output layer of the model but this is just about the actual anatomy. I think the output layer part should be moved from here.

Figure 2D: The "Overall" plot is potentially misleading — this is presumably just combining the trials from allocentric and egocentric but it is clear that the effects are driven by the allocentric portion, and it isn't really clear what the reader should conclude from overall performance when the comparison that matters for the primary points of the paper is allo v. ego. So I don't think this (and honestly most of the overall plots) should be included. Relatedly on line 124, it is misleading to say that hcDRQN yields the best performance on both tasks as hcDRQN and hcDQN are essentially equivalent on the egocentric task.

Figure 2E and lines 134-138: Can the authors demonstrate this effect statistically by, for example, sorting trials according to their distance from each switch point and then examining the extent to which performance drops (and subsequently recovers)? I also think the "overall" plot is misleading and unnecessary again here because the effects appear to still be primarily driven by the allocentric task.

Lines 162-163: There is still clear separation of strategies for the hcDQN, it just isn't linear due to the reversal that occurs at around 0 seconds. So I don't quite think it is fully accurate to say that hcDQN "mixes" strategies -- at almost all timepoints, it seems like a two-way classifier would likely be able to differentiate between these. It's also interesting that this reversal occurs, perhaps the authors can provide some intuition for why?

Figure 4A: I think it would be easier to parse these figures and compare across them if a black outline was provided around all of the possible states (this also goes for all following figures that follow a similar format).

Lines 217-218: Can the authors provide some inference for why the effect reverses for allocentric north v south?

Figure 5A: I found this figure quite difficult to parse, in part because the caption says it is for the north subtask, the main text (line 227) says both allocentric tasks, and then the figure itself is labeled Allo North, Allo South. After some effort I put together that it is indeed both subtasks being shown, but I think the task representation should be made more consistent with the representation in Figure 4A, which discretized the space similarly.

Lines 245-246: I found the writing in this section difficult to follow. The performance of hcDQN decreases relative to what? When there is 100% cue probability? According to figure 2D, hcDQN performs at this level on both tasks when the cue is always present. So I don't really follow what "decrease" is happening here. The next sentence should also be re-written ("high degree of removal 90%").

Figure 6B: Again, I think showing the overall plot here is confusing and not necessary. The primary point is that this affects the allocentric task and that the egocentric task is unaffected (because using the cues is not necessary).

Figure 6C and lines 250-252: It isn't clear to me what this analysis adds to the overall argument, so I think the authors need to motivate it more clearly. It seems fairly trivial to me that training on an environment that is more similar to the test environment would increase performance.

Figure 7D and lines 270-279: Why is cue removal 50% of the time detrimental to the egocentric task (in at least the hcDQN and maybe also the hcDRQN model), but both models are unaffected by the absence of a cue (which is what I would predict given the rest of the paper, unless I am missing something). I'm not quite sure how to interpret this, perhaps the authors can explain?

Lastly, I also found a few typos, which I think indicates that the paper could use some closer proofreading:

- Line 115: Typo: "for as a strong control" → "as a strong control"
- Line 146: "continuously rely" → "continuously relying"

- Line 152: "are in align with" → "align with" or "are in alignment with"
- Line 153: "partially environments" → "partially observable environments"
- Figure 7A: The caption does not include distractor in the list

(Remarks on code availability)

I have only briefly looked at the repository and a few of the primary files, but the README is reasonably well documented and the code appears easy to read. I have not tried to install and run the code.

Reviewer #3

(Remarks to the Author)

This manuscript presents a computational model of hippocampal function during navigation tasks. The authors develop a deep reinforcement learning agent incorporating a recurrent "CA3-like" neural network layer to investigate how the hippocampus supports goal-driven behavior in environments with limited visibility. Through a series of simulations comparing recurrent (hcDRQN) and feedforward (hcDQN) architectures, the authors demonstrate that recurrence enables the agent to successfully navigate both egocentric and allocentric tasks when sensory information is incomplete. The recurrent model not only outperforms the feedforward model in task performance but also shows greater robustness to environmental perturbations such as cue removal, maze lengthening, and sensory noise. Additionally, the authors analyze neural activity patterns from their model, showing some correspondence with experimental recordings from rodent CA1 neurons through dimensionality reduction techniques. The work suggests that the recurrent architecture of hippocampal CA3 may have evolved specifically to support navigation under realistic conditions where complete sensory information is not continuously available. While I find the modeling is technically sound and offering interesting behavioral and neural insights, several substantive issues limit the study's contribution:

1 Established principles reiterated. The importance of recurrent neural networks for processing incomplete sensory information and maintaining working memory has been a fundamental concept in both theoretical neuroscience and machine learning for decades. The idea that recurrence facilitates memory retrieval from partial cues is widely established, and applying this principle to hippocampal inspired navigation is already familiar territory in computational neuroscience. I acknowledge that the authors implement this concept effectively, the core mechanism (using recurrence to maintain a representation of cues that are no longer visible) represents an application of established principles rather than a novel computational advance. The current work demonstrates this familiar principle in a specific T maze experimental setup but doesn't really introduce fundamentally new computational mechanisms or conceptual frameworks that would significantly advance our understanding of hippocampal function beyond existing theories. This is my key concern of the presented results.

2 Another high level observation is that, the authors compared model's activity to neural activity and the neural alignment analysis remains at a coarse level. Specifically, the comparison between model activity and CA1 recordings remain only at the low-dimensional level through demixed PCA. this approach did capture broader patterns (such as strategy and temporal components), it doesn't test whether the model reproduces finer properties of hippocampal activity that are well-documented in the literature, such as place field locality (the 'shapes') and remapping properties, temporal firing rate dynamics, or the detailed structure of neural manifolds. Recent studies of hippocampal representations (Chen..Sejnowski, 2024, Levenstein..Richards 2024 and many others) have developed more granular metrics for testing computational models against neural data. A more detailed comparison across multiple scales of analysis would significantly strengthen the authors' claims about their model's biological relevance and the specific contribution of CA3 recurrence to hippocampal function.

3 Minor methodological inconsistency regarding hippocampal replay: one methodological choice worth noting is the decision to disable experience replay based on the claim that there is " lack of evidence of replay from the hippocampus to itself." This assertion overlooks established research showing that sharp wave ripple events originate in CA3, propagate to CA1, and replay experiential sequences during rest and planning (eg Buzsaki 2015 Jadhav et al. 2012). While this is a relatively minor issue that doesn't invalidate the paper's main findings, it does create an interesting contrast where a biologically documented mechanism (replay) is omitted while a biologically implausible learning algorithm (backpropagation through time) is retained. This inconsistency in applying biological constraints makes it somewhat difficult to isolate precisely which aspects of the model's performance can be attributed specifically to the recurrent architecture vs other design choices.

(Remarks on code availability)

the codebase looks appropriate.

Version 2:

Reviewer comments:

Reviewer #1

(Remarks to the Author)

Thanks to the authors for addressing the points I raised during the first round of reviews.

My overall comment was that it should be made more clear what neuroscientists can learn from this model. In my opinion, this has been sufficiently addressed by the added text clarifications. As for my major comments:

(1) about the feedforward/recurrent distinction being not as interesting as the partial observability / full observability distinction: The authors addressed this by making that latter distinction more prominent in figure 4.
(2) about block inference being side-stepped: I'm glad to see the authors have added a discussion about this.
(3) re length generalization: the authors removed the confusing phrasing about hcDQN generalization. I'm generally happy with the new interpretation, except that I would ask to remove the anthropomorphic reference to the model's "understanding" of the task.

(Remarks on code availability)

The code base looks appropriate and sufficiently documented.

Reviewer #2

(Remarks to the Author)

The authors have generally addressed each of my points and I believe the manuscript has been improved and makes an interesting and novel contribution to the literature. I have only one comment I would like the authors to revise, which is:

- Figure 2G right subpanels and related analysis: I appreciate this new analysis, but I don't think that the revised text on this point includes enough context to understand what drop and recovery mean in this plot. I think the authors should add a line explaining what "drop" and "recovery" are defined as in figure caption -- this can really just be something similar to how it was explained in the response letter.

(Remarks on code availability)

I have not run the code but it is all available and seems reasonably readable.

Reviewer #3

(Remarks to the Author)

The authors have submitted a substantially revised and improved manuscript. My previous concerns about the study's novelty and the depth of the neural analysis have been fully addressed in this new version.

The paper now does a much better job of framing its contribution, highlighting how the synthesis of reinforcement learning and partial observability provides a powerful framework for explaining goal-directed hippocampal function. Most importantly, the addition of new, granular analyses on spatial coding and place fields provides the an informative link to neural data that was previously missing. I am now satisfied that the manuscript meets the standard for publication.

(Remarks on code availability)

I did not run the code, but the code seem to be well formatted and documented.

Hippocampus facilitates reinforcement learning under partial observability

Response to reviewers

Dear Reviewers,

We would like to thank all the reviewers for the positive and constructive feedback. We have now addressed all your points in detail. In the revised manuscript we have:

1. Added a **new figure** (Fig. 5) and **updated 6 main figures** (Figs. 1-4 and 6-7) following suggestions by the reviewers;
2. Added new **3 supplementary figures** (Figs S17, S18, S19)
3. Rewritten *substantial parts* of the manuscript, including introduction, main results and discussion sections;
4. Added new supplementary table (Table 3)

Overall, we believe that our manuscript has been greatly improved in the process.

Best regards,
Dabal Pedomonti, PhD
Samia Mohinta,
Martin V. Dimitrov,
Hugo Malagon-Vina, PhD
Stephane Ciochi, PhD
Rui Ponte Costa, PhD

Reviewer #1

The authors present a model of the role of the hippocampus in ego- and allocentric navigation tasks, and compare the model's behaviour and internal representations to that of rodents in a previously recorded dataset. The model consists of a deep Q-network (DQN), and the authors systematically vary aspects of the model's architecture and inputs, such as whether there is a recurrent layer, and whether the model has access to a full top-down view of the maze or a restricted partial view. The authors find that the recurrent model, but not the feedforward model, successfully learns to navigate in both allo- and egocentric conditions. Their analysis of the model's internal representations reveals that the recurrent model has similar low-dimensional representations to those found in the animal data, and the authors show that the recurrent structure of the network enables better generalisation to different task conditions.

Major comments

This paper addresses an interesting question, of how the brain manages to optimize navigation behaviour under partial observability: not all information needed for a decision is present in the current observation. This means there's a need for some kind of working memory, which is naturally modelled with a recurrent neural network. The modelling approach is impressive and interesting and the authors did a good job linking the model to neural data, but there are a few points in the paper where I think the phrasing is confusing, making it harder to understand what is really happening. Most importantly, it would be great to flesh out a bit more what neuroscientists should learn from these AI models.

Reply: Thank you for the encouraging feedback and the useful suggestions that you provide below. We have rephrased several sections and added new panels and figures in response to your points and those of the other two reviewers. By addressing all these points we think that both the motivation and what neuroscientists can learn from our work has become more clear.

The main issue I have is that comparing the two models (recurrent and nonrecurrent) in their fit to animal data seems somewhat unfair. The way that the RL environment is set up, it would be impossible for a feedforward network to learn the allocentric task from partial view, because not all information needed for a correct decision is present in the input at any given time. For a fair comparison, I would expect to see two models that can both solve the task to a similar degree that animals do. For example, between the recurrent model with partial view and a feedforward model with full view, such as the one shown in figure S7.

Reply: Thank you. We agree that this is an important comparison. Indeed, in Fig. 3 we already use the neural activity from Fig. S7 to compare different models with partial and full observability (Fig. 3c; reproduced here for your ease). As you can see in Fig. 3c we show that models trained in fully observable

environments, including the feedforward model, do not explain the neural dynamics as well. Moreover, in Fig. S12, we compare the behaviour of models trained with full observability. Also here we find that the RNN-based model performs better. To make this point more clear we have now added a more direct comparison between partial and full observability models for behaviour alone at the end of panels c,d in Figure 4 (new panels in Fig. 4c,d), which we discuss in lines 229-234. We also show that fully observable models do not generalise as well in Fig. 7 g,h (now in Fig. 8).

In addition, we now also include a new figure (Figure 5) looking at spatial coding, and show that the RNN trained under partial observability achieves the highest spatial coding:

New Figure 5: **Recurrent agents achieve higher spatial information in partially observable environments.** (a) Example place fields for hcDRQN and hcDQN neurons with the highest, median and lowest spatial information. (b) Spatial information distributions for all CA1 model neurons across hcDRQN and hcDQN for both the baseline environment (left) and environment with longer maze (right).

Secondly, the key difficulty about the behavioural task presented in this paper is inferring from the block which task (egocentric or allocentric navigation) must be completed at the current trial. The authors side-step this issue in their modelling efforts by assuming two different output heads for the allocentric and egocentric tasks (main paper), or by providing the task ID (fig S2). That is fine, but since that information is not available to the rodents I think it would be good to include a discussion on how the brain solves this blocked multi-task learning problem. For example, some authors have suggested that the brain employs methods for finding orthogonal representations to avoid catastrophic. Other work has suggested that the brain might switch between discrete representations corresponding to different tasks, using some Bayesian inference mechanism. It would be great if the authors could comment on how they think this would be achieved.

Reply: Thank you for this suggestion. We agree that other solutions are likely to be more biologically plausible and would be important to discuss and have now included a new section on possible solutions, namely orthogonal representations and latent task inference as used in Bayesian inference frameworks (lines 379-386).

Finally, the presentation of the corridor length generalization in fig 7 is confusing. The authors write that the recurrent model “is very robust across a large range of lengths, whereas hcDQN defaults to chance level”. However, hcDQN was already at chance level for the allocentric task, so this is not a test of (failed) generalisation but of failed learning. If anything, for the egocentric task that both models can learn, the results show that the feedforward model shows better generalization to longer corridors. This makes intuitive

sense to me, because it learns a feedforward mapping from the inputs to a policy: “if I’m in a corridor with nothing else, run forward” and “if I’m at a decision point turn right”. Such a policy will generalise, whereas the RNN dynamics, when trained on a single corridor length, will be specific to that corridor length.

Reply: Thank you for pointing out this source of confusion. We agree and have now rephrased the text to make this more clear. We should highlight that when we introduce a distractor, the hcDQN agent drops performance dramatically in egocentric (now in Fig. 8e). This shows that, as pointed out by the reviewer, the DQN simply learns a mapping between states and the policy. This goes against experimental observations showing that egocentric tasks are less impacted by sensory cues (now in Fig. 8c) and the general idea that egocentric navigation should be more robust to the exact sensory information. We have rephrased this section as follows (lines 294-300):

“Our results show that hcDRQN is robust to different maze lengths in both allocentric and egocentric trials. This demonstrates that indeed the hcDRQN has successfully learned to integrate cue information, which is then maintained in its recurrent memory for action selection when required. The hcDQN is able to generalise within the egocentric task it has learned, but this is primarily due to its ability to map specific states directly to actions, rather than through any understanding of the task or memory of previous states. This limitation is evident in its failure to generalise to trials involving distractors, as shown in the distractor condition below.”

Minor comments

- Line 58: at this point it’s unclear what a “hippocampal-like” network means.

Reply: Thank you. We have removed the term hippocampal-like in this part of the text.

- Line 75-76: “on a given trial rodents have to infer the task (ego versus allocentric) through trial and error.” This is confusing. I think the authors mean “in a given block [...]” since the rule inference depends on the block.

Reply: Thank you for pointing this out; that is correct. We have updated the text.

- Line 77-78: quibble with the phrasing here: “In such setups, and in most navigational task setups rats are unlikely to be continuously accessing the full environment, implying a degree of partial observability (28)”. In RL, full observability does not necessarily mean continuous access to the whole environment. It means the agent can access all information that is relevant to decision making.

Reply: We agree, we have now rephrase this as suggested:

“In such setups -- and in most navigational task settings -- rats are unlikely to have access to all information necessary for decision-making, implying a degree of partial observability.”

- Supplementary Fig S1: I do not understand the argument that the change in head direction shows that the task is partially observable. It seems to me that the partial observability stems from the decision rule having to be inferred by the animal.

o Please annotate what is shown in these plots. The rightmost plot shows “difference”, between what and what? And what are the units?

Reply: Our interpretation of these results is as follows: in the initial and final corridors, rats increase their speed (Fig. S1a) while exhibiting fewer head turns (Fig. S1b). This suggests that they do not have continuous access to all the information required to solve the task. Instead, it appears they rely on memory of sensory cues placed in the external walls, moving quickly through the corridors and pausing to make a decision upon reaching the choice point. We apologise for the previous lack of annotations in these plots, which have been updated; they are normalised values (z-score), hence unitless, and we have clarified that the comparison is between speed (panel a) and head direction changes (panel b). These results highlight that speed is higher and head movement is reduced in the corridors, whereas the opposite pattern occurs at decision points.

- Fig 1B. According to the caption the significance stars here correspond to a “paired t-test”. I think this should be a one-sample t-test if what’s reported is the difference to chance level (50%)

Reply: We thank the reviewer for this helpful comment. The reviewer is correct, as the appropriate test is a one-sample t-test against chance level (50%), and we have now corrected this in the caption.

- 117-118: “We did not use replay in our model because of the lack of evidence of replay from the hippocampus to itself”. Can the authors clarify what is meant by this? There is ample evidence for replay in the hippocampus. While many theoretical models assume that these are read out by cortex during consolidation, I am not aware of any more evidence for that than for read-out by the hippocampus itself. If this exists, please provide the references.

Reply: Thank you. We did not need replay in our setup. Also since replay is commonly framed in terms of hippocampal-to-cortical systems consolidation, we felt it more appropriate not to include it; we have now rephrased this part as follows (lines 116-120):

“Unlike standard DQN implementations, our setting did not require experience replay (Mnih et al., 2015). We therefore simplified our model by omitting this component. While experience replay is often interpreted in terms of hippocampal–cortical systems consolidation, there is also evidence for internal hippocampal replay (Jadhav et al., 2012; Buzsáki et al., 2015). Replay mechanisms are likely essential for stable reinforcement learning in more complex tasks than the ones that we consider (Mnih et al., 2015).”

- 153: “trained in partially environments”, missing “observable”

Reply: Thank you. We fixed it now.

- 157-170: the dPCA analysis needs more explanation. How was the given label of each component identified? Further, the authors write that “both hcDRQN and hcDQN exhibit a temporally decaying population dynamics [...] These features of the hcDRQN highlight its unique ability to integrate input information over time ...”. If both the hcDQN and the hcDRQN show this temporal decay, how does that highlight the hcDRQN’s unique ability?

Reply: Thank you for spotting! We have clarified this as follows:

Next, we find that both hcDRQN and hcDQN exhibit temporally decaying population dynamics (Fig. 3a middle). However, whereas the hcDRQN starts flat and then decays, the hcDQN increases before it decays. These hcDRQN neural dynamics are more in line with the animal neural dynamics (cf. Fig. 3c).

- Fig 6. What's the point of this figure? It makes sense in machine learning, but what can neuroscientists learn about this?

Reply: We have improved this section with motivation and implications from a neuroscience stand point.

We now start with the following motivation:

“both biological and artificial recurrent neural networks can also integrate evidence over time in probabilistic tasks Hanks et al. 2015, Singer et al. 2021, Pemberton et al. 2024)”

And finish by stating the following:

“Finally, when analysing trajectory behaviour, our model shows that agents trained with probabilistic cues spend more time at the starting location to integrate sensory evidence for longer before committing to a decision (Fig. 7d). This result is in line with experimental observations showing that rats integrate sensory evidence over time and take longer to respond when this evidence is more ambiguous (Brunton et al. 2013).

Taken together, these results suggest that CA3 recurrence plays a critical role in learning to navigate stochastic environments, consistent with the broader importance of recurrent dynamics for sensory integration in the brain (Brunton et al. 2013, Mante et al. 2013, Hanks et al. 2015, Pemberton et al. 2024). ”

Reviewer #2

In this work, Pedamonti et al. demonstrate that including recurrence in a hippocampus-like neural network architecture allows agents to learn to associate past stimuli with future states, which is necessary for adaptive behavior in partially observable environments. This also gives rise to a series of behaviors and neural population responses that are similar to rats in a simple but classic task with (the plus maze with allocentric navigation and location-based cues) and without (the plus maze with egocentric navigation) task-relevant partial observability. Several extensions to the environment are also thoroughly explored. Overall, I enjoyed this paper and found the results to be largely sensible and important. I think the work will be interesting to many researchers in this space. I do however have a few points that I would like the authors to address in a revision.

Reply: We would like to thank the reviewer for the positive and constructive feedback. We have addressed the suggestions/points raised in detail as explained below.

First and most importantly, while I am convinced by the paper that recurrence is needed to associate past sensory cues with future states, I am less convinced that it is specifically CA3 recurrence that is required to accomplish this. This is because CA3 recurrence (via the trisynaptic pathway) is not the only type of recurrence in the hippocampal formation that can solve this sort of problem. “Big loop” recurrence via the monosynaptic pathway (e.g. bidirectional links between EC and CA1) has been shown to allow statistical learning, which yields another mechanism for the hippocampus to link past sensory cues with future states, and thus to solve the problem of partial observability (e.g. Schapiro et al., 2017). There is indeed no mention of this work in the current paper, and I think at minimum the authors should discuss this possibility. Claims throughout about CA3 recurrence being the primary hippocampal mechanism at play here need to be weakened unless the authors can show that a model with only CA3 recurrence captures this behavior better than a model with only a monosynaptic pathway. This seems particularly important because Anna Schapiro’s line of work demonstrates that the trisynaptic pathway actually fails to learn regularities across individual episodes in the presence of big-loop recurrence via the monosynaptic pathway. A large part of why this occurs is also because there are several other biologically-relevant traits of CA3 that the present work doesn’t explicitly account for, namely high inhibition and resulting sparse connections. Fully addressing this point would require accounting for these traits of CA3 as well as more explicit modeling of the EC, which I recognize is a substantial departure from the current model. But I think it is important to acknowledge that this abstraction is a major weakness of the current work relative to state-of-the-art models of the hippocampus. Ideally, this point should be addressed by comparing to an architecture that is more biologically-realistic and accounts for this possibility. If such modeling is not feasible, these weaknesses should be acknowledged with writing and framing overhauls throughout the paper.

Reply: Thank you for raising this important point. We fully agree that other recurrent loops within the hippocampal formation—particularly the EC–CA1 loop—may also contribute to the types of computations examined in our study, but were not included in our model. Our initial aim was to explore whether a parsimonious model focusing on CA3 recurrence could be sufficient to support behaviour under partial observability. However, we do not claim that CA3

recurrence is the sole mechanism at play, and we now recognise the importance of acknowledging the potential contributions of other pathways.

At this point, it would be a substantial amount of work for us to include models with other forms of recurrence. Instead, and following your suggestions, we have revised the manuscript to tone down claims about the specific role of CA3 recurrence, and to explicitly discuss the relevance of work by Anna Schapiro and others on statistical learning via EC–CA1 “big-loop” recurrence. We apologise for not including this important literature in the original version. Indeed, we had suspected that related work existed that we had not fully incorporated, and your comments have helped us address that gap.

Discussion: We have now added a new discussion on this point:

“While here we have focused on a CA3-like recurrence in guiding learning to navigate partially observable environments, we acknowledge that CA3 is not the only source of recurrence within the hippocampal formation capable of supporting such computations. Notably, prior work by Schapiro and others (Schapiro et al. 2013, Schapiro et al. 2017, Koster et al. 2018) has demonstrated that “big-loop” recurrence—mediated by bidirectional connections between entorhinal cortex (EC) and CA1 via the monosynaptic pathway—can also support statistical learning and temporal integration. This circuit provides an alternative mechanism by which the hippocampus might associate past sensory cues with future states, especially in scenarios requiring the accumulation of information across episodes. Future work should consider the multiple hippocampal loops, which can then be evaluated in light of the framework that we introduce here, but also that of others (Schapiro et al. 2013, Schapiro et al. 2017, Koster et al. 2018).”

Toned down role of CA3:

We have toned down the claims of a CA3 role throughout the manuscript. Including, explicitly mentioning other forms of hippocampal recurrence in the introduction:

“Other studies have shown that the hippocampus, and the hippocampal recurrence, such as that provided by CA3 and CA1-EC loops, is involved in integrating information over time as required for navigational tasks where sensory cues are no longer present (Lee et al. 2002, Lee et al. 2003, Gilbert et al. 2006, Schapiro et al. 2013, Schapiro et al. 2017, Koster et al. 2018)”

In the main text describing our Fig. 2 results, we have also toned down the role of CA3, for example, the final paragraph now reads as follows:

“Given that animals are unlikely to have continuous access to the full environment in most realistic settings, our results suggest that hippocampal recurrence—such as that provided by CA3, though not exclusively—plays an important role in supporting goal-directed behaviour under naturalistic conditions.”

We have also removed references to CA3 in several places, instead referring to *hippocampal recurrence*, as suggested by the reviewer.

I have several other line and figure-specific comments:

- Line 18: The way this sentence is written in the abstract makes it sound like the agents are performing some kind of dimensionality reduction themselves, but I believe that the authors mean they use dimensionality reduction techniques to investigate this. This should be made more clear.

Reply: We thank the reviewer for this helpful observation. We agree that the original wording may suggest that the agents themselves performed dimensionality reduction, which was not intended. We have revised the sentence in the abstract to clarify that dimensionality reduction was applied to the agents' internal representations in our analysis:

“Next, we used dimensionality reduction of the agents' internal representations to extract components reflecting reward, strategy, and temporal representations, which we validated experimentally using hippocampal recordings.”

- Lines 40-42: The writing here makes it sound like all model-based approaches are successor representations (SR), which is of course not true, so that should also be made more clear.

Reply: We thank the reviewer for pointing this out. We agree that the original wording may incorrectly suggest that. We have revised the text to clarify that successor representation is a hybrid approach:

“Another line of research has shown that some model-based approaches, such as Successor Representation, can explain anticipatory features that have been observed experimentally (Dayan et al. 1993, Stachenfeld et al. 2017).”

- Lines 47-48: It is not clear from this brief mention why a hierarchical model accelerates learning, and why it is important to mention this.

Reply: We thank the reviewer for this comment. We agree that this sentence did not provide sufficient context and was not essential to the main narrative. We have therefore removed this statement from the revised manuscript.

Supplementary Figure S1: What is the difference plot? It was not clear to me what the y-axis represents here and it is not mentioned in the figure caption.

Reply: We have fixed this. The difference plot now clarifies that it is the difference between the two previous plots (so speed minus head direction change). We have also improved the captions.

Figure 1A: From the results main text alone, it isn't clear if the sensory cue locations directly mimic those from the experimental setup. Maybe it'd help to represent the cue locations somehow in Fig 1A?

Reply: We thank the reviewer for the suggestion. We have updated Figure 1a to include a schematic of visual landmarks to show the presence of distal sensory cues. However, note that we cannot guarantee that their placement does not exactly match the location utilised in our simulated environment. However, all of our analysis suggests that indeed our cue placement is able to capture a number of neural and behavioural features.

Figure 1B: Are these box plots over individual trials? Would it be more informative to see this broken out by individual subjects? Perhaps a violin plot in the background over individual

trials (if this is indeed what the box plots represent) to give a sense of the distribution of trials with individual animal means plotted on top would be more informative?

Reply: We thank the reviewer for the suggestion. We have updated Figure 1b to display violin plots, with individual animal means overlaid on top (also reproduced below). Note that we were not plotting performance on individual trials before, so this new plot is consistent with the previous plot. Also note that here we use 8 animals instead of only 5 which we use elsewhere in our paper as we have behavioural data from different studies: 5 from Ciochi et al. [1] and 3 from Malagon-Vina et al. [2].

[1] Ciochi et al., *Science* 348, 560–563 (2015)

[2] Malagon-Vina et al., *Nat. Commun.* 9, 309 (2018)

Figure 2A: The caption mentions the output layer of the model but this is just about the actual anatomy. I think the output layer part should be moved from here.

Reply: Thank you for pointing that out, we have removed it from the caption (a).

Figure 2D: The "Overall" plot is potentially misleading — this is presumably just combining the trials from allocentric and egocentric but it is clear that the effects are driven by the allocentric portion, and it isn't really clear what the reader should conclude from overall performance when the comparison that matters for the primary points of the paper is allo v. ego. So I don't think this (and honestly most of the overall plots) should be included. Relatedly on line 124, it is misleading to say that hcDRQN yields the best performance on both tasks as hcDRQN and hcDQN are essentially equivalent on the egocentric task.

Reply: We thank the reviewer for these suggestions. We have now updated Figure 2 by removing all of the "Overall" plots. We have also corrected the statement in line 124 (new line 126). We would like to keep some of the overall plots in other figures as they help provide a sense of the overall task performance.

Figure 2E and lines 134-138: Can the authors demonstrate this effect statistically by, for example, sorting trials according to their distance from each switch point and then examining the extent to which performance drops (and subsequently recovers)? I also think the "overall" plot is misleading and unnecessary again here because the effects appear to still be primarily driven by the allocentric task.

Reply: Thank you for this suggestion. In the revised manuscript we now align every trial to the precise block-switch point and quantify both the immediate performance drop and the subsequent recovery. For each block, trials were re-indexed so that trial 0 corresponds to the first trial after the switch; the first twenty trials of every block were then averaged across all

blocks and agents. We define the “drop” as the difference between the mean performance of the first block and the minimum of the remaining blocks, and the “recovery” as the difference between the mean of the last two blocks (plateau values) and the minimum calculated in the previous drop. When comparing these metrics for the allocentric blocks, the drop and recovery effects remained significant ($p_{\text{drop}} = 1.18\text{e-}5$, $p_{\text{rec}} = 0.002$), with hcDQN having larger delta values. Neither drop nor recovery reached significance in the egocentric task, confirming that switching is difficult for the allocentric condition. To present these results more clearly, we removed the previous “overall” subplot, as suggested, and replaced it with four sub-panels (Figure 2g): the left subpanels show the trial-aligned mean \pm s.e.m. curves for allocentric and egocentric blocks, while the right subpanels display the corresponding drop and recovery in performance. This new panel is also reproduced below:

The relevant text has been updated accordingly to highlight the allocentric-specific drop and rapid recovery of hcDRQN:

“Performance aligned to allocentric goal switches shows that hcDRQN has small delta values for the drops and recovery, reflecting lower errors to changes in goal location (Fig. 2g, top row). hcDQN, by contrast, shows larger delta values, indicating a failure to form or update allocentric representations. Comparing hcDRQN vs hcDQN, both the drop and recovery effects remained significant ($p_{\text{drop}} = 1.18\text{e-}5$, $p_{\text{rec}} = 0.002$) in the allocentric case, confirming that hcDRQN learns and adapts better to the allocentric subtask. No such effects are observed in the egocentric task (Fig. 2g, bottom row).”

Lines 162-163: There is still clear separation of strategies for the hcDQN, it just isn't linear due to the reversal that occurs at around 0 seconds. So I don't quite think it is fully accurate to say that hcDQN "mixes" strategies -- at almost all timepoints, it seems like a two-way classifier would likely be able to differentiate between these. It's also interesting that this reversal occurs, perhaps the authors can provide some intuition for why?

Reply: We agree with the reviewer that our original phrasing was unclear. What we meant is that a linear decoder can clearly separate strategies in the animal and hcDRQN, but not in hcDQN, as now more clearly shown in the new Fig. 3b panels (also on the right here). While strategy-related activity is still present in hcDQN, it does not align along a consistent linear axis. We have now rephrased this text for improved clarity as:

“First, we find that the population dynamics for allo and egocentric tasks can be linearly separable when using the hcDRQN strategy demixed PCs, but not for hcDQN (Fig. 3a left, Fig. 3b). This reversal between tasks by hcDQN is likely due to its inability to learn both allocentric and egocentric tasks (Fig. 2).”

Figure 4A: I think it would be easier to parse these figures and compare across them if a black outline was provided around all of the possible states (this also goes for all following figures that follow a similar format).

Reply: We thank the reviewer for this helpful suggestion! We have now updated Figure 4a and all similar figures to include a black outline around all possible states.

Lines 217-218: Can the authors provide some inference for why the effect reverses for allocentric north v south?

Reply: We thank the reviewer for this insightful question. The apparent reversal effect arises due to how performance is measured relative to the correct goal location. In the allocentric condition, the hcDQN model consistently selects the wrong goal location on the exact opposite arm of the T-maze. However, our error metric assumes that the agent is targeting the correct goal. This means that if the agent reliably chooses the wrong location, the computed distance error may still appear smaller in some conditions (e.g., allocentric north vs. south) due to maze geometry, despite the agent fundamentally failing the task.

We have clarified this point in the revised manuscript to make the interpretation more transparent and to note that this does not reflect successful behaviour by the hcDQN:

“Our results show that hcDRQN outperforms hcDQN, except for a minor reversal in the allocentric north vs. south condition. This is explained by the fact that the hcDQN consistently selects the wrong goal location; however, our error metric assumes the agent is targeting the correct goal. As a result, choosing the opposite arm can, due to maze geometry, yield a smaller distance error in some comparisons even though the agent fails the task.”

Figure 5A: I found this figure quite difficult to parse, in part because the caption says it is for the north subtask, the main text (line 227) says both allocentric tasks, and then the figure itself is labeled Allo North, Allo South. After some effort I put together that it is indeed both subtasks being shown, but I think the task representation should be made more consistent with the representation in Figure 4A, which discretized the space similarly.

Reply: We thank the reviewer for pointing this out. We have now removed the reference to the “north subtask” from the caption, updated the figure by adding grid lines to match the discretisation style used in Figure 4a. Moreover, we added state numbers to match with Figure 5b state representations (now in Fig. 6a).

Lines 245-246: I found the writing in this section difficult to follow. The performance of hcDQN decreases relative to what? When there is 100% cue probability? According to figure 2D, hcDQN performs at this level on both tasks when the cue is always present. So I don't really follow what "decrease" is happening here. The next sentence should also be re-written ("high degree of removal 90%").

Reply: We thank the reviewer for highlighting this confusion. We agree that the wording was unclear. In the original text, “performance decreases” was meant to contrast hcDQN with hcDRQN, but as the reviewer correctly notes, the performance of hcDQN remains flat across

all levels of cue availability. We have now revised the paragraph to clarify that hcDQN does not adapt its behaviour based on cue probability and its performance remains constant because it follows a fixed policy that does not rely on cues. In contrast, the hcDRQN adapts its behaviour based on cue probability, showing a gradual decline in performance as cue availability decreases. This highlights its dependency on input integration.

We have rephrased this section in the manuscript (lines 272-276):

“During inference, the performance of hcDRQN gradually declines as cue availability decreases, reflecting its dependency on sensory cue integration (Fig. 7a). In contrast, hcDQN shows no change in performance across different cue probabilities (Fig. 7b), indicating that it does not rely on cue input and instead follows a fixed policy. Notably, hcDRQN maintains performance above 60% even at 90% cue removal, demonstrating robustness to reduced sensory input.”

Figure 6B: Again, I think showing the overall plot here is confusing and not necessary. The primary point is that this affects the allocentric task and that the egocentric task is unaffected (because using the cues is not necessary).

Reply: We thank the reviewer for this helpful suggestion. We have now removed the overall plot and enhanced the heatmaps by adding a grid overlay to make individual states more distinguishable.

Figure 6C and lines 250-252: It isn't clear to me what this analysis adds to the overall argument, so I think the authors need to motivate it more clearly. It seems fairly trivial to me that training on an environment that is more similar to the test environment would increase performance.

Reply: We thank the reviewer for raising this point. Our manuscript focuses on testing models against environments with partial observability. A natural step from what we have done until this point is to also train models with stochastic cues, as it's likely to happen in more realistic settings. We agree that this result is intuitive, but it's useful to confirm that RNN-based models can also learn to integrate information in stochastic environments. As a consequence of stochastic training, we observe that the hcDRQN spends more time in the initial position, which makes a specific prediction that can be tested experimentally. Indeed, this prediction seems in line with the observation that under higher ambiguity, animals wait for longer to make a decision (Brunton et al. 2013, Mukherjee, Lam et al. Nature 2021).

While the result may appear intuitive, we included this analysis to confirm that the models were not simply memorising the maze layout or a fixed policy during training. By explicitly testing models trained with and without stochastic cue removal, we were able to verify that performance improves in test environments with similar levels of cue removal. We have clarified this motivation in the revised manuscript:

“Next, we trained the hcDRQN in a stochastic environment, where cues are randomly removed (Fig. 7c). This analysis helped to confirm that, as expected, RNN-based models can learn to integrate in stochastic environments. Indeed, we observed that training with stochastic cue removal led to more robust performance during testing (~10% improvement). Interestingly, when analysing trajectory behaviour, our model shows that agents trained with probabilistic cues spend more time at the starting location to integrate sensory evidence for

longer before committing to a decision (Fig. 7d). This result is in line with experimental observations showing that rats integrate sensory evidence over time and take longer to respond when this evidence is more ambiguous (Brunton et al. 2013, Mukherjee et al. 2021).”

Figure 7D and lines 270-279: Why is cue removal 50% of the time detrimental to the egocentric task (in at least the hcDQN and maybe also the hcDRQN model), but both models are unaffected by the absence of a cue (which is what I would predict given the rest of the paper, unless I am missing something). I'm not quite sure how to interpret this, perhaps the authors can explain?

Reply: We thank the reviewer for raising this point. This test condition differs from the earlier cue removal experiments. Here, we systematically control cue availability across three fixed conditions: (i) both cues always present (100%), (ii) both cues always absent (0%), and (iii) only one cue present (50% condition, i.e., half of the cues are present in each trial).

1. In the 100% condition, the agents perform well as this matches the training distribution.
2. In the 0% condition, the models default to a baseline strategy, which is similar to their behaviour when traversing the central part of the corridor (i.e., lacking cues near the starting point).
3. However, the 50% condition represents a novel scenario that the models never experienced during training. As a result, both models exhibit impaired performance in this untrained configuration.

We have clarified how this in the text as follows:

“Next, we tested the models on the same environment it was trained on, but removing a set of cues at a time. Note that this is a complete removal of cues, rather than stochastic cues as in the previous section.”

Lastly, I also found a few typos, which I think indicates that the paper could use some closer proofreading:

- Line 115: Typo: "for as a strong control" → "as a strong control"
- Line 146: "continuously rely" → "continuously relying"
- Line 152: "are in align with" → "align with" or "are in alignment with"
- Line 153: "partially environments" → "partially observable environments"
- Figure 7A: The caption does not include distractor in the list

Reply: We thank the reviewer for carefully identifying these issues. We have corrected all of the mentioned typos and updated the caption of Figure 7a (now Fig. 8a) to include the distractor condition. We have also carefully proofread the entire manuscript to ensure clarity and correctness.

Reviewer #3

This manuscript presents a computational model of hippocampal function during navigation tasks. The authors develop a deep reinforcement learning agent incorporating a recurrent "CA3-like" neural network layer to investigate how the hippocampus supports goal-driven behavior in environments with limited visibility. Through a series of simulations comparing recurrent (hcDRQN) and feedforward (hcDQN) architectures, the authors demonstrate that recurrence enables the agent to successfully navigate both egocentric and allocentric tasks when sensory information is incomplete. The recurrent model not only outperforms the feedforward model in task performance but also shows greater robustness to environmental perturbations such as cue removal, maze lengthening, and sensory noise. Additionally, the authors analyze neural activity patterns from their model, showing some correspondence with experimental recordings from rodent CA1 neurons through dimensionality reduction techniques. The work suggests that the recurrent architecture of hippocampal CA3 may have evolved specifically to support navigation under realistic conditions where complete sensory information is not continuously available. While I find the modeling is technically sound and offering interesting behavioral and neural insights, several substantive issues limit the study's contribution:

1 Established principles reiterated. The importance of recurrent neural networks for processing incomplete sensory information and maintaining working memory has been a fundamental concept in both theoretical neuroscience and machine learning for decades. The idea that recurrence facilitates memory retrieval from partial cues is widely established, and applying this principle to hippocampal inspired navigation is already familiar territory in computational neuroscience. I acknowledge that the authors implement this concept effectively, the core mechanism (using recurrence to maintain a representation of cues that are no longer visible) represents an application of established principles rather than a novel computational advance. The current work demonstrates this familiar principle in a specific T maze experimental setup but doesn't really introduce fundamentally new computational mechanisms or conceptual frameworks that would significantly advance our understanding of hippocampal function beyond existing theories. This is my key concern of the presented results.

Reply: Thank you for the thoughtful feedback. We agree that the role of recurrence in maintaining representations of past sensory input is well established in neuroscience. Indeed, this was a key motivation for our work – to understand if this principle enables us to better explain hippocampal-dependent behaviour and representations. As far as we know this topic and its importance for goal-driven tasks under partial observability have not been explicitly studied in the hippocampal field. We think that the novelty in our study is to demonstrate its importance through a novel combination of deep RL, RNNs, multi-tasking, behaviour and neural dynamics (including new experimental results):

- 1. Training deep RL models with partially observable environments is needed to account for hippocampal observations:** We show that training deep reinforcement learning (RL) agents with recurrent architectures to navigate environments in a goal-directed manner yields neural representations that better align with experimental data. Specifically, our agents develop reward- and task-specific representations that

closely match those observed experimentally (Figs. 1, 2, 3, 4, 8). Crucially, training under partial observability is key to achieving this alignment. While recurrence is well established as essential for handling partial observability, to our knowledge, the integration of RL, recurrence, and direct comparison with experimental observations has not been thoroughly explored within the hippocampal modelling literature.

- 2. RL models are sufficient to account for goal-driven features in the hippocampus:** We have focused on a deep RL framework, which contrasts with the majority of hippocampal models that typically employ RNNs to predict the next observation or position (e.g., Cueva & Wei, 2018; Recanatesi et al., 2021; Schaeffer et al., 2022; Levenstein et al., bioRxiv), rather than to learn task outcomes within an RL framework as we do here. Our findings demonstrate that learning solely through task outcomes — i.e., via reinforcement signals — is sufficient to explain both behavioural and neural patterns in the data we analysed. Moreover, we train our models in a multi-task setting, which allows us to emphasise the importance of hippocampal representations and partial observability. This is another element that is not commonly seen in hippocampal models.

This does not preclude the importance of self-supervised learning (SSL). On the contrary, we think that the hippocampus likely integrates both RL and SSL mechanisms. While RL captures goal-directed behaviour and dynamics, as shown in our work, incorporating SSL may account for additional, well-established features of hippocampal spatial coding identified in previous studies. We have now updated our discussion to make these points more explicit (lines 393-403).

- 3. Novel experimental observations:** One of our contributions is the close interaction between modelling and (novel) experimental observations. In particular, the (demixed) dimensionality reduction analysis that we performed on our neural data identified behaviourally relevant population dynamics regarding rule, decision and time (Fig. 3), which to the best of our knowledge are novel and lend support to the notion that hippocampal representations encode goal-driven behavioural variables (Fig. 3). Moreover, our work suggests the hippocampus learns a common latent space that can be used for multiple goal-driven tasks, ego and allocentric in our case, and that RL can be sufficient to achieve this.

In summary, while our work does build on established new principles, we believe these findings offer new insight into the computational role of hippocampal recurrence and highlight important considerations of partial observability for both modeling and experimental design.

2 Another high level observation is that, the authors compared model's activity to neural activity and the neural alignment analysis remains at a coarse level. Specifically, the comparison between model activity and CA1 recordings remain only at the low-dimensional level through demixed PCA. this approach did capture broader patterns (such as strategy and temporal components), it doesn't test whether the model reproduces finer properties of hippocampal activity that are well-documented in the literature, such as place field locality (the 'shapes') and remapping properties, temporal firing rate dynamics, or the detailed

structure of neural manifolds. Recent studies of hippocampal representations (Chen..Sejnowski, 2024, Levenstein..Richards 2024 and many others) have developed more granular metrics for testing computational models against neural data. A more detailed comparison across multiple scales of analysis would significantly strengthen the authors' claims about their model's biological relevance and the specific contribution of CA3 recurrence to hippocampal function.

Reply: We thank the reviewer for this suggestion. Although it is not our focus here to try to reproduce existing hippocampal observations of spatial coding, we do agree that we should analyse spatial coding in our agents. Following the reviewers suggestion, in the revised manuscript, we have now added a new main Figure (**new Figure 5**) and a number of new supplementary figures (**new Figs. S17, S18 and S19**) in which we provide a number of new comparisons using both standard spatial metrics and looking at how the low dimensional representations of the model encode space. We detail these new analyses below.

However, we should clarify that our goal here is not to try to explain a wide range of classical observations, but instead focus on the role of partial observability and that it is needed to obtain several desirable features including better explaining behaviour and neural dynamics in the context of our own task and data. We do agree that the natural next step for our work is to consider how this generalises to settings such as those by Chen..Sejnowski, 2024, Levenstein..Richards 2024, and study in more precise detail the importance of partial observability across a wide range of tasks and conditions. We now more clearly state this in the discussion (lines 393-403).

New spatial analysis

New Figure 5: Recurrent agents achieve higher spatial information in partially observable environments. (a) Example place fields for hcDRQN and hcDQN neurons with the highest, median and lowest spatial information. (b) Spatial information distributions for all CA1 model neurons across hcDRQN and hcDQN for both the baseline environment (left) and environment with longer maze (right).

We have now included a new main figure (reproduced above) in which we use spatial information, as suggested by the reviewer. Our results show that hcDRQN achieves higher spatial information, as shown by the example place maps developed (Fig. 5a; more examples in the new Fig. S17) and corresponding spatial information (Fig. 5b). Importantly, spatial information drops substantially when trained in fully observable environments. These distributions align closely with values reported in dorsal CA1 (Markus et. al, 1994), reinforcing the biological plausibility of our recurrent architecture. These results reinforce our key point that partial observability is an important feature to consider when modelling hippocampal observations.

We also included other metrics such as sparsity index and spatial coherence (Figure S18, Table S3). The hcDRQN model achieves significantly higher spatial information (as shown in Fig. 5b) and lower sparsity (indicating more selective and efficient spatial coding) compared to the other models. Moreover, we visualised population activity using PCA across different task contexts (Figure S19). The hcDRQN model exhibits smooth, continuous trajectories with clear spatial structure by position, reward, and step number. Interestingly, when colour-coding by starting location (North vs South), hcDRQN shows a shared latent structure, suggesting that it has learnt a good task structure (see below). In contrast, hcDQN shows a strongly overlapping structure, whereas the full-view hcDRQN and hcDQN show a strong separation, suggesting that none of these models learnt a good task structure, which is in line with our results throughout the paper.

Together, we believe that these additions strengthen our claims regarding the joint role of recurrence and partial observability in providing a richer model of hippocampal function.

3 Minor methodological inconsistency regarding hippocampal replay: one methodological choice worth noting is the decision to disable experience replay based on the claim that there is "lack of evidence of replay from the hippocampus to itself." This assertion overlooks established research showing that sharp wave ripple events originate in CA3, propagate to CA1, and replay experiential sequences during rest and planning (eg Buzsaki 2015 Jadhav et al. 2012). While this is a relatively minor issue that doesn't invalidate the paper's main findings, it does create an interesting contrast where a biologically documented mechanism (replay) is omitted while a biologically implausible learning algorithm (backpropagation through time) is retained. This inconsistency in applying biological constraints makes it somewhat difficult to isolate precisely which aspects of the model's performance can be attributed specifically to the recurrent architecture vs other design choices.

Reply: Thank you for raising this. Our reading of the literature is that the classical view is that of replay as a means of hippocampal-cortical systems consolidation. However, the primary reason for not including it was because we did not need it for our relatively simple RL setup. We have now rephrased this part of the text to acknowledge that there evidence for "internal" replay, referring to the studies highlighted by the reviewer as follows (new lines 116-120):

"Unlike standard DQN implementations, our setting did not require experience replay (Mnih et al., 2015). We therefore simplified our model by omitting this component. While experience replay is often interpreted in terms of hippocampal-cortical systems consolidation, there is also evidence for internal hippocampal replay (Jadhav et al., 2012; Buzsáki et al., 2015). Replay mechanisms are likely essential for stable reinforcement learning in more complex tasks than the ones that we consider (Mnih et al., 2015)."

Hippocampus facilitates reinforcement learning under partial observability

Response to reviewers

Dear Editor,

We are happy to see that all the reviewers are happy with our comprehensive revisions. We have addressed all the remaining minor points by R1 and R2 as indicated below.

Reviewer #1 (Remarks to the Author):

Thanks to the authors for addressing the points I raised during the first round of reviews.

My overall comment was that it should be made more clear what neuroscientists can learn from this model. In my opinion, this has been sufficiently addressed by the added text clarifications.

As for my major comments:

(1) about the feedforward/recurrent distinction being not as interesting as the partial observability / full observability distinction: The authors addressed this by making that latter distinction more prominent in figure 4.

Reply: Thank you.

(2) about block inference being side-stepped: I'm glad to see the authors have added a discussion about this.

Reply: Thank you.

(3) re length generalization: the authors removed the confusing phrasing about hcDQN generalization. I'm generally happy with the new interpretation, except that I would ask to remove the anthropomorphic reference to the model's "understanding" of the task.

Reply: We have reworded this part as: "rather than through a **model** of the task or memory of previous states.", where it previously read as: "rather than through any **understanding** of the task or memory of previous states."

Reviewer #1 (Remarks on code availability):

The code base looks appropriate and sufficiently documented.

Reviewer #2 (Remarks to the Author):

The authors have generally addressed each of my points and I believe the manuscript has been improved and makes an interesting and novel contribution to the literature. I have only one comment I would like the authors to revise, which is:

- Figure 2G right subpanels and related analysis: I appreciate this new analysis, but I don't think that the revised text on this point includes enough context to understand what drop and recovery mean in this plot. I think the authors should add a line explaining what "drop" and "recovery" are defined as in figure caption -- this can really just be something similar to how it was explained in the response letter.

Reply: Thank you for pointing out this lack of clarity. As suggested by the reviewer we have added the following text used in the rebuttal to the main text: Drop is defined as the difference between the mean performance of the first block and the minimum of the remaining blocks, and the recovery as the difference between the mean of the last two blocks (plateau values) and the minimum calculated in the previous drop.

Reviewer #2 (Remarks on code availability):

I have not run the code but it is all available and seems reasonably readable.

Reviewer #3 (Remarks to the Author):

The authors have submitted a substantially revised and improved manuscript. My previous concerns about the study's novelty and the depth of the neural analysis have been fully addressed in this new version.

The paper now does a much better job of framing its contribution, highlighting how the synthesis of reinforcement learning and partial observability provides a powerful framework for explaining goal-directed hippocampal function. Most importantly, the addition of new, granular analyses on spatial coding and place fields provides the an informative link to neural data that was previously missing. I am now satisfied that the manuscript meets the standard for publication.

Reply: Thank you!

Reviewer #3 (Remarks on code availability):

I did not run the code, but the code seem to be well formatted and documented.